# Diurnal variability of the Atmospheric Boundary Layer height over a tropical station in the Indian Monsoon Region

Sanjay Kumar Mehta[1], Madineni Venkat Ratnam[2], Sukumarapillai V. Sunilkumar[3], Daggumati Narayana Rao[1], and Boddapaty V. Krishna Murthy[1]

[1]SRM Research Institute, SRM University, Kattankulathur-603203, India
[2]National Atmospheric Research Laboratory, Gadanki-517112, India
[3]Space Physics Laboratory (SPL), VSSC, Trivandrum-695022, India

*Correspondence to*: Sanjay Kumar Mehta (sanjaykumar.r@res.srmuniv.ac.in; ksanjaym@gmail.com)

**Abstract.** The diurnal variation of atmospheric boundary layer (ABL) height is studied using high resolution radiosonde observations available at every 3-h intervals for 3 days continuously from 34 intensive campaigns conducted during the period December 2010-March 2014 over a tropical station Gadanki (13.5$^{o}$N, 79.2$^{o}$E, 375 m), in the Indian monsoon region. The heights of the ABL during the different stages of its diurnal evolution, namely, the convective boundary layer (CBL), the stable boundary layer (SBL), and the residual layer (RL) are obtained to study the diurnal variability. A clear diurnal variation in 9 campaigns is observed while in 7 campaigns the SBL does not form for the entire day and in the remaining 18 campaigns the SBL form intermittently. The SBL forms for 33%-55% of the time during nighttime and 9% and 25% during the evening and morning hours, respectively. The mean SBL height is within 0.3 km above the surface which increases slightly just after midnight (0200 IST) and remains almost constant till morning. The mean CBL height is within 3.0 km above the surface which generally increases from morning to evening. The mean RL height is within 2 km above the surface which generally decreases slowly as the night progresses. The diurnal variation of the ABL height over the Indian region is stronger during the pre-monsoon and weaker during winter season. The CBL is higher during the summer monsoon and lower during the winter season while the RL is higher during the winter season and lower during the summer season. During all seasons, the ABL height peaks during the afternoon (~1400 IST) and remains elevated till evening (~1700 IST). The ABL suddenly collapses at 2000 IST and increases slightly over night. Interestingly, it is found that the low level clouds have an effect on the ABL height variability, but not the deep convective clouds. The lifting condensation level (LCL) is generally found to occur below the ABL for the majority of the database and they are randomly related.

## 1 Introduction

The atmospheric boundary layer (ABL) is the lowest layer of the troposphere in which the flow field is directly influenced by the interaction of the Earth's surface at a response time scale of about an hour or less (Stull 1988, Garratt, 1994). The importance of the ABL stems from the fact that it is the gateway for the pollutants and anthropogenic emissions, moisture, heat and momentum fluxes to the free atmosphere. Rapid transport in the ABL takes place in order to achieve the radiative balance between the surface and the free atmosphere. The ABL is the largest sink for atmospheric kinetic energy. The ABL height is a key parameters, providing a length

scale for the vertical extent and concentration of atmospheric pollutants, convective activity, and cloud and fog formation (Deardorff, 1972; Holtslag and Nieuwstadt, 1986; Seibert et al., 2000; Konor et al., 2009). The diurnal variability is a dominant feature of the ABL, which plays an important role in the exchanges of heat, momentum, moisture, and chemical constituents between the surface and free atmosphere.

Depending on the physical process, the diurnal pattern of the ABL is mainly classified into three major layers: the convective boundary layer (CBL), the stable boundary layer (SBL), and the residual layer (RL) (Stull, 1988; Garratt, 1994).The CBL evolves during the daytime just after the sunrise due to convective turbulence (thermals of warmer air) associated with entrainment zone, a stable layer, on its top. Just before the sunset and during night, as the thermals cease to develop, the CBL collapses and the SBL form due to fast cooling of the surface. Even though thermals cease to develop during nighttime the mean state variables remain nearly the same as the former CBL, creating the RL associated with capping inversion layers, a stable layer, on its top. Thus, the RL is disconnected from the ground by underlying SBL having no source of turbulence generation for its maintenance–rather turbulence decays homogeneously in all directions. One aspect of the RL that has been pointed out as being important for the diurnal evolution of the ABL is that it provides the potential for "explosive growth" of the ABL as a CBL forms in the morning and grows into the RL. Generally, the mean ABL height lies between 0.03-3.0 km above the surface (Stull, 1988). However, the CBL can be as high as ~ 5.0 km during a midsummer day in low-latitude deserts and as low as ~0.5 km over the ocean. The SBL height generally is less than 0.5 km above the surface (Garratt, 1992).

As daytime the surface flux (the convectively-induced turbulence) due to solar heating is stronger than night-time surface flux (wind-induced turbulence) due to surface friction causes diurnal variation in the ABL. However, diurnal variability of the ABL is not only determined by the variabilities in the surface fluxes, but also on meteorological conditions such as the presence of the clouds, horizontal advection of air and subsidence from aloft. These forcings considerably affect turbulence development and the growth of the ABL. As, for example, the presence of clouds greatly influences the turbulent structure due to local radiative heating or cooling. The diurnal variation is generally strong, mainly over land in the absence of any upper level cloud and at the time of year when surface temperatures are the highest (Angevine et al., 2001).

The ABL height is often determined from the vertical profiles of temperature, humidity, and wind components obtained from radiosonde measurements (Schmid and Niyogi, 2012). Most of the routine radiosondes only operate twice a day at 0000 and 1200 UTC, which are not suited well to study the diurnal variations of the ABL height. Thus, there have been only a few studies on diurnal variability of the ABL height (Angevine et al., 2001) mainly due to non-availability of the required measurements with adequate time resolution. High resolution radiosondes launched at sufficiently close time intervals (less than 1-h) can provide direct information on diurnal variability of ABL height. Liu and Liang (2010) studied the climatology of the ABL height diurnal cycle, using fine-resolution soundings launched at intervals ranging from 1-12 hours collected in 14 major field campaigns around the world. They found a strong diurnal cycle, both over land and oceans, whereas the cycle is weak over ice. Seidel et al., (2010) using routine radiosondes over the globe found significant differences in day and night ABL heights. In an another study, Seidel et al., (2012) reported the seasonal pattern in the diurnal cycle of the ABL using 3 hourly ERA-Interim data and 6 hourly climate models outputs over the continental United States and Europe. There are several case studies focusing on the diurnal structure of the boundary layer and the mechanism responsible for its formation over different regions of the globe. Brill and Albrecht, (1982)

presented the diurnal variation of the cloud fraction and trade-wind inversion base height using the data collected from various ships and aircraft. May et al., (2012) have studied the diurnal variation of convection, cloud, radiation, and boundary layer structure in the coastal monsoon environment (Darwin, Australia). Santanello et al., (2007) has studied the feedback of soil moisture dryness on the development of the convective boundary layer over the southern Atmospheric Research Measurement Program–Great Plains Cloud and Radiation Testbed (ARM-CART) sites. During a clear sky day Hashiguchi et al., (1995a) observed boundary layer radar echo indicating the ABL height. Hashiguchi et al., (1995b) further observed that the boundary layer radar detects diurnal variation of the ABL both at the equatorial and midlatitude regions. During the summer over a midlatitude glacier, the diurnal and vertical and horizontal structures of the boundary layer were found to be dominated by persistent glacier winds forced by gravity (Van den Broeke, 1997).

Recently, various remote sensing systems such as lidar (Tucker et al., 2009), sodar (Shravan Kumar and Anandan, 2009), wind profiler (Kumar and Jain, 2006; Bianco et al., 2011), Radio Acoustic Sounding System (RASS) (Clifford et al., 1994; Chandrasekhar Sarma et al., 2008), ceilometer (van der Kamp and McKendry, 2010) have been developed for continuous direct measurements or estimates to study the diurnal variation of the ABL height (Seibert et al., 2000). Sodar can generally provide the SBL height but not always the CBL height due to inadequate height coverage. The lidars make use of aerosol extinction profiles and can provide information on diurnal variability of the ABL height. Using network of wind profilers located in California's Central Valley, Bianco et al. (2011) studied the diurnal evolution of the CBL which attains maximum height 3-4 h before the sunset. A few studies on the different ABL regimes (CBL and SBL) and their evening transition have been carried out using various remote sensing instruments located at Gadanki (Kumar and Jain, 2006; Basha and Ratnam, 2009; Kumar et al., 2012; Sandeep et al., 2015).These studies show that the mean CBL height is within 3.5 km and the mean SBL height lies below 0.6 km above the surface and their transition from the CBL to the SBL occurs about one and half hour before the sunset. However, the complete diurnal variation of the ABL height has not been reported either using single instrument or a combination of two or more in the above mentioned studies.

Over Gadanki (13.45$^o$N, 79.2$^o$E), a tropical location in the Indian monsoon region high resolution GPS radiosonde launches were carried out in 3-h intervals for three consecutive days in each month during the period December 2010- March 2014. Making use of this dataset, first time complete diurnal variability of the ABL height and their classification into different ABL regimes such as the CBL, SBL and RL during different seasons and effect of the cloud in its diurnal structure has been studied and the results are presented in this paper. Section 2 describes the data and methodology, results are presented in section 3, and in section 4 discussion and concluding remarks are presented.

**2 Data and Method of analysis**

**2.1 GPS Radiosonde data**

As part of Tropical Tropopause Dynamics (TTD) experiment under Climate and Weather of Sun-Earth System-India (CAWSES-India) program, intensive campaigns of high resolution GPS radiosonde were conducted to study the diurnal variability of the ABL over a tropical station, Gadanki, located at 375 m above mean sea level. The radiosondes were launched at 3-h intervals for 3 consecutive days in each month from December 2010 to

March 2014, except during March 2011, December 2012, January-February 2013 and April 2013. Each campaign started at 1100 IST (=UTC+0530) on day one and ended at 0800 IST on the day four. Table 1 shows the dates of radiosonde launchings. Most of the observations are conducted during non-rainy days except two during 01:00 IST-02:00 IST on 18 August 2011 and 14:00 IST-20:00 IST on 21 August 2012, with total rainfall about 47 mm and 46 mm, respectively. In total 764 profiles of temperature, pressure, relative humidity and

horizontal wind are obtained from the 34 campaigns. These data are collected using 'Meisei RD-06G' radiosonde observations sampled at 10 m (sampled at 2 Sec intervals) under the TTD campaigns (Ratnam et al., 2014). The observed data set is gridded uniformly to 30 m altitude resolution interval so as to remove any outliers arising from random motions or very high frequency fluctuations but the same time to retain the ABL signature. Note that gridding these data to coarser resolution (e.g. 100 m) smooths out the ABL detection,

especially the SBL, which lies generally below 0.5 km above ground level. Quality checks are then applied to remove any further outliers arising due to various reasons to ensure high quality in the data (Mehta et al., 2011).

**2.2 Infrared Brightness Temperature (TBB) data**

In order to understand the role of clouds especially low-level clouds occurring around the ABL cloud top temperature (CTH) is estimated using Infrared Brightness Temperature (TBB) obtained from the Climate

Prediction Centre, NOAA which available at a time resolution of one hour and at a spatial resolution of $0.03^{o}$ X $0.03^{o}$.. This is globally-merged, full-resolution (~4 km) IR data formed from the ~11 micron IR channels aboard the GMS-5, GOES-8, Goes-10, Meteosat-7 and Meteosat-5 geostationary satellites. The data have been corrected for zenith angle dependence to reduce discontinuities between adjacent geostationary satellites. For this study, we averaged the TBB data into $0.25^{o}$ latitude X $0.25^{o}$ longitude around Gadanki and collected for

every three hours during each campaign. The cloud top height is obtained as altitude corresponding to averaged TBB from radiosonde temperature profiles.

**2.3 Method of analysis**

Altitude profiles of temperature variables and moisture variables obtained from radiosonde observations are used to estimate the ABL height based on different methods. Seven different methods, two using the

temperature profile, three using the moisture profile and two a combination of temperature and moisture are adopted to estimate the ABL height in each sounding. The temperature variables are dry air temperature ($T$), potential temperature ($\theta$) and moisture variables are relative humidity (RH), specific humidity ($q$) water vapor pressure ($P_w$). A combination of both temperature and moisture variables are virtual potential temperature ($\theta_v$) and radio refractivity ($N$). The ABL height is generally identified as the location of (1) the maximum vertical

gradient of one of the variables: $T$, $\theta$, and $\theta_v$ or (2) the minimum vertical gradient one of the variables: RH, $q$, $P_w$ , and $N$ (Sokolovskiy et al., 2006; Basha and Ratnam, 2009; Seidel et al., 2010; Chan and Wood, 2013) below 3.5 km above the surface. We limited the altitude region to 3.5 km, following Chan and Wood (2013) who used GPS radio occultation refractivity data to study the seasonal cycle of the ABL over the globe. The upper limit 3.5 km is selected in order to avoid the midlevel inversions, if any. When more than one peak in the

gradient occurs below 3.5 km, the lowest peak having a value greater than 80% of the main peak is considered as the ABL top. As suggested by Ao et al. (2012), the gradient based ABL definitions are most meaningful when they are large in magnitude relative to the average gradient. They defined the "sharpness parameter" as

$X'_S = |-X'_{min \, or \, max}/X'_{RMS}|$ where $X$ is moisture or temperature variables and $X'_{RMS}$ is the root-mean square (RMS) value of $X'$ over the altitude range 0-3.5 km. If $X'_S \geq 1.25$ it is considered that ABL is well defined. In the case of SBL the gradient at the top of the residual layer is much stronger than that of the gradient at the SBL. The SBL is generally identified using surface based inversion methods (Seidel et al., 2010). At the SBL, where temperature increases sharply, the temperature gradient shows a maximum value immediately just above the surface, but not at the actual level where the temperature reverses from the positive to negative gradient. In the present study, the SBL is identified as the level of maximum temperature below 0.9km. The upper limit for the SBL height identification is based on Kumar et al. (2012), who observed the maximum wind speed (sporadic region) deep enough up to~0.9 km. The method to obtain the RL is similar to that of the CBL.

Following the above criteria, we have obtained the CBL, SBL and RL heights during each campaign listed in Table 1. Out of 764 profiles, 17 profiles are rejected due to bad data quality. In the night time two types of profiles are observed; one in which the SBL is present and other in which the SBL is not present. The profiles for which the SBL is defined, are further subdivided into two cases; i) with the RL not defined and ii) the RL defined. As the observations are at 3-h intervals, actual changes happening during the morning transition (MT) and evening transitions (ET) during the course of diurnal cycle might not have been captured. Sandeep et al. (2015) have made a comprehensive study on the transitory nature of the ABL during ET over Gadanki. They found that the transition follows a top-to-bottom evolution. Note that the mean sunrise time is about 0545 IST (0630 IST) while sunset time is about 1830 IST (1745 IST) during the summer (winter) over Gadanki.

The lifting condensation level (LCL) is defined as the height at which an unsaturated air parcel becomes saturated (RH >100%) when it is cooled by dry adiabatic lifting (Wallace and Hobbs, 2006). It provides an empirical estimate of the cloud base height. The temperature ($T_L$) pressure ($P_L$) and height ($Z_L$) of the LCL is obtained using following equations (Bolton 1980; Stull 1988; Anurose et al., 2016):

$$T_L = \frac{2840}{3.5 \ln(T_{30m}) - \ln(Pw_{30m}) - 4.805} + 55 \qquad (1)$$

$$P_L = P_{30m} \left[ \frac{T_L}{T_{30m}} \right]^{3.5} \qquad (2)$$

$$Z_L = -H \ln(P_L/P_0) \qquad (3)$$

where $T_{30m}$, $P_{30m}$ and $Pw_{30m}$ are temperature, pressure and water vapor pressure at 30 m height, respectively, $P_0$ is surface pressure and $H$ is scale height taken as 7.5 km (Wallace and Hobbs 2006).

## 3 Results

### 3.1 Topography, balloon trajectory and mean wind pattern over Gadanki

Figure 1a shows the seasonal mean trajectories of balloon below 4 km and locations of balloons at 4 km during different seasons launched over Gadanki for the period December 2010-March 2014. Gadanki is surrounded by hills of maximum height of 500-600 m above the mean sea level. Towards west of the Gadanki is the chain of the Nallamala hills with height about 800-1000 m and east side of it is flanked by Bay of Bengal. Note that Gadanki is a far inland station (~120 km away from the Bay of Bengal coast) and hence does not have any local effect due to the sea. The balloon generally has drifted about $0.1^o$ and $0.2^o$ in latitude and longitude, respectively below 4 km. It can be seen that balloon ascends almost vertically and has drifted towards the southwest

direction during the winter season (DJF; December- January-February) while during summer monsoon season (JJA; June-July-August) they are drifted mostly towards southeast direction. During pre-monsoon (MAM; March-April-May) balloon ascends almost vertically up to 1 km and drifts southward above up to 4 km. In the post-monsoon months (SON; September-October-November) it ascends almost vertically up to 2-3 km and change direction towards southeast.

Figures 1b-c show the seasonal mean zonal ($U$) and meridional ($V$) winds, respectively, obtained using averaging horizontal wind data observed during the period December 2010- March 2014. Within the ABL, it can be seen that the zonal winds are mostly easterly during the winter and pre-monsoon while westerly during the summer monsoon and post monsoon and the meridional winds remain southerly throughout the year. During the summer monsoon season, low level westerly jet (LLJ) core of 10 m/s at 1 km is clearly evident.

**3.2 Identification of the ABL (CBL and SBL) from temperature and moisture profiles**

Figure 2 shows the typical profiles of the temperature variables $T$, $\theta$, $\theta_v$ and the moisture variables RH, $q$, $P_w$ and $N$ observed at 1100 IST on 8 February 2011 to identify the CBL. Note that these profiles are observed in clear sky conditions (TBB ~295 K). It can be seen that the CBL is capped by the inversion layer of thickness 0.150 km where a sharp increase in temperature and a sharp decrease in the moisture variables occur. The base of the entrainment zone is at 0.69 km above the ground level, which is defined as the top of the CBL. Within the CBL, $\theta$, $\theta_v$, and $q$ and $P_w$ are almost constant with altitude signifying that the air is mixed or having a tendency towards vertical mixing due to the action of the turbulence, a characteristic of the 'mixing layer' (Seibert et al., 2000). Above the CBL and within the entrainment zone $T$, $\theta$ and $\theta_v$ increase sharply by about 1.5 K, 3.0 K and 2.0 K, respectively, and moisture variables (RH, $q$, $P_w$ and $N$) decrease sharply by about 6 times. At the top of the entrainment zone, $\theta_v$ coincides with $\theta$ as water vapor concentration becomes very small. These sharp changes at the CBL top are easily captured by the gradient of the temperature and moisture variables as shown in Figs. 2b and 2d, respectively. Both the maximum gradient of the temperature variables and the minimum gradient of the moisture variables identify the CBL height quite well.

As mentioned earlier, after sunset (in the nighttime) the identification of the SBL is not as easy as the identification of the CBL, mainly because of the absence or delay in the formation of the surface inversion due to weak surface cooling. Therefore, whenever the SBL is not present the ABL is represented as the RL. Typical examples of identification of the SBL are shown in Figs. 3a-3l for three different types of nighttime temperature ($T$, $\theta$ and $\theta_v$) and moisture (RH, $q$, $P_w$ and $N$) profiles and their corresponding gradient profiles indicating the presence of the SBL but not the RL, both the SBL and the RL, not the SBL but the RL, respectively. The temperature and moisture profiles and their corresponding gradient profiles observed at 0200 IST on February 9, 2011 show the evolution of the SBL only but not the RL (Figs. 3a-3d). The gradients of temperature profiles and moisture profiles shown in Fig. 3b and Fig. 3d are offset by scale 10. These profiles are observed during fair weather conditions (TBB ~ 295 K). The top of the SBL identified based on the surface based the inversion (SBI) in the $T$ is indicated as a horizontal dashed line in Fig 3a. The SBL is identified at 0.39 km above the ground level. Within the SBL both $\theta$ and $\theta_v$ increase steeply. The temperature gradient profiles show a maximum gradient at the surface which steeply decreases within the SBL (Fig 3b). The moisture profiles (Fig. 3c) also show a steep decrease within the SBL. However, their gradients (Fig. 3d) show a negative peak at 0.18 km observed at a lower height when compared to top of the SBL. The negative gradient peaks in the moisture

variables are denoted as open circles. These sharp changes in the moisture variables in the SBL are indicative of inversion forming adjacent to the surface similar to the formation of entrainment zone aloft in the development of the CBL. Thus, one can also identify the height of the SBL based on moisture gradient peaks (especially when temperature observation is not available) near the surface when the RL is absent. However, it becomes difficult when the gradient aloft is strong as will be seen in the later examples.

A similar feature can be noticed in the case of the SBL forming beneath the RL as identified using temperature and moisture variables, shown in Figs. 3e-3h observed at 0200 IST on December 19, 2013. These profiles are observed under deep convective case with the TBB value about 262.3 K and corresponding CTH is 6.67 km. In this case, the SBL based on the SBI is identified at 0.21 km above the ground level (Fig. 3e). The temperature variables increase and moisture variables decrease in the SBL (Figs 3e and 3g), but not as steep as previous profiles shown in Figs. 3a and 3c, respectively. The gradient method identifies the RL top at 1.35 km denoted by open circles as shown in Figs. 3f and 3h. The RL has similar features as the CBL mentioned in Fig 2. One can observe a weak gradient in the moisture variables (Fig. 3h) at 0.30 km. It can be assigned as the top of the SBL defined using the gradient method in the moisture, which is slightly higher than the SBL defined using surface based inversion. However, identification of the negative gradient peak in the moisture profiles becomes difficult when strong RL is present as can be noticed from the next example.

Unlike the above mentioned cases, an example indicating no SBL formation beneath the RL in both temperature and moisture profiles observed at 0200 IST on February 26, 2014 is shown Figs. 3i-3l. In this case, the feature of the RL is similar to that shown in Fig. 2 for the CBL and that shown in Figs. 3e-3h for the RL. Like CBL, it also starts from the surface. By definition the RL is the layer observed above the SBL and hence not considered as the ABL. However, if the SBL is absent as in this case, the RL will be above the surface and it is nothing but nighttime ABL. In a study more similar in concept to ours, Liu and Liang (2010) pointed that such cases are generally identified with near-neutral conditions in the surface layer which they assigned as the neutral RL (NRL) that starts from the ground surface. However, in our study, we refer to it as the RL. Using gradient method, the top of the RL is identified at 1.47 km from the temperature variables and at 1.14 km from the moisture variables. The moisture profiles and its corresponding gradients are disturbed for about 1.0 km above the RL. Unlike the previous examples, the moisture profiles in this case do not sharply decrease. The TBB at 0200 IST on February 26, 2014 is 289.4 K and corresponding CTH of 0.81 km, indicating the presence of the low level fair weather clouds. The RL height difference observed using temperature and moisture variables is difficult to explain, but seems related to the effect of the cloud. The surface moisture is very small and the LCL observed at about 0.6 km indicates that a shallow layer cloud of thickness about 0.2 km decoupled from the surface (Garratt, 1992). The absence of the SBL indicates lack of sufficient surface cooling which could be due to presence of cloud above radiatively warming the surface. It is to be noted that there is a weak gradient present in both temperature and moisture variables at 0.51 km which is very small when compared to the gradient at the RL. Comparing the moisture gradients below the RL top shown in Fig. 3l with that presented in Fig. 3h, it is inferred that the gradient below the RL may not always be due to surface cooling. Hence, identifying the SBL using moisture variables based gradient method needs caution. Thus, in the present study, we preferred the identification of the SBL based on the SBI only.

**3.3 Typical diurnal variation of the ABL**

From the previous typical examples presented for particular times, it is observed that the ABL identified using temperature and moisture variables independently agree fairly well. Thus, any one temperature variable and any one moisture variable are sufficient to document the ABL variation. It is also observed that the ABL heights are well identified during different cloudy conditions. Thus, hereafter we only examine $T$, $\theta_v$, $q$ and $N$ for the analysis. $\theta$ and RH and $P_w$ are dropped from further analysis as they are related to $\theta_v$ and $q$, respectively. The typical examples of the three hourly variations of the CBL, SBL, RL, CTH and the LCL observed over three days are shown in Figs. 4a-4d for four different types of diurnal variation indicating formations of well-defined SBL, delayed SBL, no SBL, and intermittent SBL, respectively. Figs. 4a-4d also show four different cloudy conditions occurring around the ABL, far above (deep convection) the ABL, below the ABL and just above the ABL, respectively. First three (Figs. 4a-4c) panels show the complete 3 days observations for which the typical examples of the particular times have been described earlier in Fig 2 and Fig 3. Note that the diurnal variability of the CBL and RL is identified based on $T$, $\theta_v$, $q$ and $N$ variables, while the SBL is identified using $T$ only.

Figure 4a shows the perfect diurnal evolution of the ABL for all the 3 days observed during February 8-11, 2012 (see supplementary Fig. S1, which depicts the corresponding vertical profiles $T$, $q$ and $N$ along with TBB and CTH). The CBL height just before noon (1100 IST) is at height 0.69 km on the first day, which further grows to height 1.2 km during afternoon (1400 IST) owing to the maximum surface heating and remains at the same height till evening (1700 IST) just before the sunset (1816 IST). During this time ET takes place and development of the SBL started at height 0.48 km at 2000 IST, which descends slowly during its nighttime evolution to the height 0.15 km at 0800 IST on the second day. Due to development of convective turbulence after the sunrise (0639 IST), the SBL is in the stage of the disappearance after 0800 IST, probably indicating the activity of MT. During the second and third days, the evolutionary feature of CBL and SBL are similar to that on the first day; however, they show large day to day variability. The CBL was not identified either by using moisture or by temperature variables at 1100 IST on second day (Fig. S1). Either clear sky conditions or the CTH below the CBL is observed during the first and third days (TBB ~ 295 K). During the second day, TBB reduced by about 10 K indicating presence of low level clouds with CTH at 2.5 km. The diurnal variation of the ABL shows the CBL at the same height using all the variables on the first day, but they differ on the second and third days. The CBL height from the minimum gradients of $q$ and $N$ are exactly same, indicating the dominance of the moisture part in the $N$ when compared to $T$. Similarly, both $T$ and $\theta_v$ identify same heights of the CBL and RL indicating dominance of temperature in $\theta_v$ when compared to moisture. The RL is not observed on the first night; it appears on the second and third nights, which generally decreases in the course of the night.

A similar example of typical 3 days diurnal variation of the ABL identified using $T$, $\theta_v$, $q$ and $N$ profiles in which the SBL formation delay observed during December 18-21, 2013 is shown Fig. 4b (see corresponding vertical profiles in Fig. S2). This case is observed during deep convection events. In this case, the CBL height just before noon is at 1.59 km and remains at about the same height till evening. After the sunset, the CBL does not collapse until midnight and the SBL has not formed, indicating delay in ET process. Thus, the ABL heights observed at 2000 IST and 2300 IST are the RL top heights. The SBL appears at height 0.18 km after midnight (0200 IST) and remains at about the same height till morning (0800 IST) on the second day. On the second day, the CBL reappears again at 1.35 km before noon, which became maximum (1.83 km) afternoon and steadily decreases to 1.53 km at 2000 IST. The ET process again delayed and the SBL did not form until just before the

midnight on the second day. On the third day, the CBL varies in similar fashion as on the second day but this time the ET process was not delayed and the SBL formed on 2000 IST. On the third night the SBL was detected at a height about 0.45 km. Note that both the temperature and moisture profiles show wave like feature during 2000 IST-0800 IST on third night could be either due to strong horizontal advection or due to gravity wave propagation (Fig S2) which will be examined in a separate study. The RL during all three days remain about the same height in contrast to previous example where it decreases as the night progresses.

The TBB lies between the range 285 K to 218 K corresponding to the CTH about 2.46 km to 11.6 km, respectively. By the first day (December 18, 2013) to the evening of the third day (December 20, 2013), deep convection prevails with the CTH above 6 km except a few occasions when it lowers down to about 3.0-4.0 km. From midnight of the third day CTH observed below 3.0 km, during which $T$, $q$ and $N$ were observed disturbed (Fig S2). The delay in the ET processes seems related to warming caused by cloudy skies, which might have resulted in a delay of the surface cooling during the early part of night of first and second day. However, it must be noted that presence of the clouds could be one possible reason as there as occasions when the SBL forms even in the presence of the clouds. Hence, the delay in the ET process cannot be entirely explained by the longwave effect. It can be seen that, during the deep convection case, the ABL identified using different moisture and temperature variables is the same.

Another typical 3 days variation of the ABL, where the SBL is not formed throughout the observation period and when the cloud lying below the ABL as shown in Fig 4c is observed during February 24-27, 2014 (See Fig S3 for vertical profiles). Thus, during nighttime only the RL is defined which is above the surface and considered as nighttime ABL. None of the temperature profiles show the evolution of the SBI (Fig S3). The ABL is at about 2.0 km (1100-1700 IST) which descends down to height 1.14 km (2000 IST) and to 0.66 km (2300 IST) following ET before midnight on the first day. The CTH indicates the presence of low level cloud at about 1.0-1.5 km since the evening of the first day to early morning of the second day. During these cloudy days, the ABL heights (~1.5 – 2.5 km) are higher than when compared to the ABL height (0.6-1.5 km) determined during clear sky days. An example of the typical 3 days variation of the ABL, where SBL is defined intermittently and when cloud lying above the ABL is shown in Fig 4d observed during May 29-June 01, 2012 (See Fig. S4 for vertical profiles). By intermittent is meant the SBL defined for a few times and not the whole night. In this case, the TBB lies between the range 290 K to 276 K corresponding to the CTH about 1.96 km to 3.60 km, respectively. The CTH is always above the ABL. On the first day, the SBL does not form and the ABL is almost at about the same height till the evening of the second day. On second and third nights the SBL formed for sometimes. In all the above cases, the SBL either has decreased after midnight or remained constant. These two types of the SBL are somewhat equivalent in types reported by Kumar et al., (2012). During clear-sky conditions, the constant and decreasing SBL height after midnight are generally accompanied by steady and unsteady winds (Kumar et al., 2012).

The typical diurnal variation of the LCL for the four different cases of the clouds is shown in Figures 4a-d. In general, it is observed that the LCL forms below the ABL (CBL+RL) except a few cases. During February, 8-11, 2011, the LCL occurs just above the CBL (Fig. 4a) but below the RL. During the second day when the CTH occur at 2.4 km, the LCL also forms near to it. However, the LCL also is higher than the CBL on the first day and the third day, when the CTH is very low. It indicates that the LCL and CTH are not always related. In the case when deep convention is observed during December, 18-21, 2013, the LCL shows nice diurnal cycle as the

ABL (CBL+SBL) (Fig. 4b). During February 24-27, 2014, the case when the CTH is occurring below the ABL, the LCL also occurs below the ABL and at or around the CTH (Fig. 4c). During May 29-June 01, 2012, when the CTH is well above the ABL, the LCL remains almost constant and varies in a similar way as the CBL and RL (Fig. 4d). From these typical cases, it appears that the ABL and the LCL are sometimes closely related and

sometimes randomly, a detail comparison of the LCL with different ABL regimes is presented in the later section.

### 3.4 Correlation analysis

The typical examples of the SBL, CBL and RL identification shown in previous section reveal that the different methods agree well except in a few cases. In order to see the overall correlation of the CBL and RL heights

detected from $T$, $\theta_v$, $q$ and $N$, a statistical comparison between them has been made as shown in Fig. 5. The total number of observations available during daytime (0800 IST -1700 IST)) is 379. Out of them, the CBL is defined in 360 profiles (see Table 1), in 18 profiles the CBL is not defined and one profile is rejected due to bad data quality. The total number of observations available during nighttime (2000 IST-0500 IST) is 385. Out of them, the RL is defined in 320 profiles as listed in Table 1, in 49 profiles the RL is not defined and 16 profiles are

rejected due to bad data quality during the quality check process. Figs. 5a-5d show the scatter plots of the CBL heights obtained using four different methods.

The correlation between the CBL heights obtained from $T$ and $\theta_v$ (r = 0.97) (Fig. 5a)  and $q$ and $N$ (r = 0.99) (Fig. 5d) are found to be excellent, which have a standard deviation (SD) of0.16 km and 0.10 km, respectively, suggesting that it can be determined using either of the methods. However, the correlation between the CBL

heights obtained from $\theta_v$ and $q$ (r = 0.79) (Fig. 5b) and $T$ and $N$ (r = 0.78) (Fig. 5c) though agreeing well, have a large SD of about 0.39 km and 0.38 km, respectively. Several times, the CBL height determined from temperature variables is higher than that one obtained from the moisture variables. There could be various reasons for this disparity, however, whenever the temperature and moisture gradients are sharper or having significant gradients (Basha and Ratnam, 2009), both methods define unique height, but differences occur when

they are not so sharp. Seidel et al., (2010) observed no correlation between the ABL heights obtained using $T$ (elevated inversion) and the rest of the methods ($\theta_v$, $q$, $N$). Surprisingly, they observed good correlation between $\theta$ and moisture variables ($q$, RH, $N$), but no correlation with $T$. In fact, $\theta$ mostly depends upon the $T$ variation, so one would expect a good correlation between them as observed in our case.

Figures 5e-5h show the scatter plots of the RL heights obtained using four different methods. Similar to the

CBL heights, the RL heights also show excellent correlation between $T$ and $\theta_v$ (r = 0.94) (Fig. 5e) and $q$ and $N$ (r = 0.96) (Fig. 5h) with SD about 0.24 km and 0.17 km, respectively. The correlation between RL heights obtained from $\theta_v$ and $q$ (r = 0.86) (Fig. 5f)  and $T$ and $N$ (r = 0.88) (Fig. 5g) with SD about 0.32 km and 0.27 km, respectively, are comparatively better than the CBL heights. However, unlike the CBL heights, the RL heights estimated using different methods scatter uniformly about the linear fit, indicating that sometimes the RL

heights obtained using temperature variables are higher than that of moisture variables and vice- versa. As we have observed excellent correlation between different methods, hereafter rest of the results will be presented using $T$ variable only, because both the SBL and CBL can be easily estimated using this variable.

### 3.5 Statistics of the SBL, CBL, and RL heights

Before proceeding to the diurnal variation of the ABL, we first document the occurrence statistics of the SBL,
CBL and RL and the general nature of their diurnal cycle as shown in Fig. 6.  The occurrence of the SBL, CBL
and RL and the occurrences of the SBL during different seasons are shown in Fig. 6a and 6b, respectively. SBL
forms mainly during nighttime, except a few times during the early evening (1700 IST) and the late morning
(0800 IST). The SBL occurrence dominates at nighttime (2300-0500 IST) with occurrence about 50%. At 2000
IST, occurrence of the SBL is only 33%, indicating that delay in surface cooling for about 17% of the times.
Over Gadanki, the occurrence of the SBL is less than the occurrence over land in the midlatitude (North
America and European regions)(Liu and Liang, 2010). The SBL appeared at 0800 IST for about 25% of the time
indicating the dominance of the surface cooling even after (generally 2 hours after) the sunrise. As surface
cooling starts well before (generally 2 hours before) the sunset, sometimes (about 9%) SBL also forms at 1700
IST. In general, the SBL occurs more frequently during the winter when compared to the summer monsoon
season.  Liu and Liang (2010) also observed occurrences of the SBL a few times during midday. However, we
have not observed such occurrence over Gadanki. It is interesting to note that SBL at 0800 IST mostly formed
during the winter months. During winter, when sunrise is at ~ 0630 IST surface cooling may remain strong till
0800 IST on some days leading to the formation of the SBL. Whereas during the summer monsoon season with
sunrise at about 0545 IST, surface cooling may not last till 0800 IST leading to very few occurrences of the SBL
at 0800 IST. The CBL and RL occurrences dominate and are evenly distributed at 3-h intervals during daytime
(0800-1700 IST) and nighttime (2000-0500 IST), respectively. In contrast to Liu and Liang, (2010), we
observed the uniform occurrence of the CBL at 3-h intervals during daytime.

Figure 6c-6e shows the mean height of the SBL, CBL and RL along with their respective standard deviations
between 1100 IST -0800 IST. The mean SBL height varies between $0.16 \pm 0.07$ km at 2000 IST to $0.29\pm 0.21$
km at 2000 IST. The overall mean SBL is below0.3 km above the surface consistent with the literature (Stull,
1998). After the sunrise, the CBL starts to form which lies between $1.4 \pm 0.62$ km at 0800 IST to ~ $2.0\pm0.5$ km
at 1400 to 1700 IST (Fig. 6b). Overall the mean CBL height is well below the 3.0 km above the surface
consistent with the literature (Stull, 1988; Garratt, 1994). The daytime CBL remains prevalent as part of the RL
during nighttime, which slightly falls to $1.8\pm0.67$ km at 2000 IST and becomes minimum $1.6\pm0.55$ km at 0200
IST (Fig. 6c).

We examine the probability distributions in order to find out the most probable height at which the SBL,
CBL and RL occur in winter, summer monsoon seasons and in the whole year as shown in Fig. 7. Figs. 7a-7c
show the SBL height distribution which has a clear peak at 0.15 km in the annual as well as during the winter
and summer are lower than the mean SBL height. Both maximum distributions and mean indicates that the SBL
heights are within the 0.30 km as consistent with the literature except a few times when SBL forms above it.
The mean SBL height shows clear seasonal variation with lower height during the summer than the winter
season. Figs. 7d-7f show the CBL height, which has clear peaks at about 2.0 km in the annual and winter season
closely coinciding with the mean CBL. During the summer season, the CBL height distribution shows a broad
peak between 0.8 km and 2.4 km, and the mean CBL height is slightly higher than that during the winter season.
A clear seasonal variation is also observed in the mean CBL height. It also indicates that CBL heights are highly
variable during summer monsoon season when compared to winter season. The RL height distribution is similar
to the CBL height distribution in the annual and winter peaking at a lower height ~ 1.8 km (Figs. 7g-7h).

During the summer monsoon season, the RL height distribution shows a peak at 0.8 km in contrast to the CBL distribution. Liu and Liang (2010) also observed similar distributions as reported in this study.

We also obtained the distribution of the SBL, CBL and RL for the annual, winter and summer monsoon in terms of the boxplot as shown in the Figs. 8a-c, respectively. The median values the SBL during the annual, winter and summer remain same (Fig. 8a). There are a few outliers whose values are greater than 3 times of the corresponding interquartile ranges (IQR) for the annual and two different seasons. The SBL mostly lies below 0.65 km during the winter and 0.4 km during summer monsoon. The SBL is more variable during the winter

than summer monsoon. As mentioned earlier, the CBL is higher and more variable during the summer monsoon than winter (Fig. 8b). Similarly, the RL is also more variable during the summer monsoon than winter (Fig. 8c). In contrast to the CBL, RL is lower during the summer monsoon when compared to winter.

**3.6 Diurnal and Seasonal Variation of the ABL height**

We have two combinations of the ABL variability; one is the CBL and the SBL denoted as the $ABL_{CS}$ and

another is the CBL and the RL denoted as the $ABL_{CR}$. The diurnal variation of the $ABL_{CS}$ height, $ABL_{CR}$ height, the surface temperature and the LCL height are shown in Fig. 9. The surface temperature is taken from the automatic weather station (AWS) observations over Gadanki. The diurnal variation of the ABL comprises of the CBL observed between 0800 IST and 1700 IST, the SBL at 1700 IST and 0800 IST and the RL observed between 2000 IST and 0500 IST. As mentioned earlier, the mean CBL varies between ~0.5 km and 3.0 km and

the mean SBL varies between ~ 0.09 km and 0.6 km. The mean diurnal variation of the CBL and the SBL ($ABL_{CS}$ hereafter) shows that the CBL evolved slowly with time attains maximum height at 1400 IST and either remains constant or decrease slowly till 1700 IST and then collapse to the SBL. The mean variation of the SBL over time is very small. It can be seen that the ET is more abrupt at 1700 IST than the morning rise at 0800 IST consistent to earlier studies (e.g. Liu and Liang, 2010).

The variation of the CBL and RL ($ABL_{CR}$ hereafter) over three days for all the campaigns is shown in Fig. 9b. Note that the CBL variation at 1700 and 0800 IST presented in Fig. 9a is relatively lower because former also includes the SBL observations; especially during 1700 IST and 0800 IST (see Fig 6). It is interesting to note that the RL falls to lower height most of the night and thus, the $ABL_{CR}$ shows a diurnal pattern with maximum height during 1700 IST and minimum during early morning 0800 IST. The observed maximum

height of the CBL at 1700 IST is found to be consistent with the general circulation model output (Medeiros et al., 2005). The RL height varies from~0.5 km to ~3.0 km. The RL is present throughout the night during most of the day, which is sustained by the presence of the relatively warm air trapped between two stable layers, RL at the top and recently developed SBL due to surface cooling at the bottom. The trapped warm air slowly becomes cooler due to exchange of heat to the adjoining free atmosphere and gradually intensifying SBL results in the

descent of the RL during the course of the night, allowing the turbulence to decrease homogeneously in all directions.

The diurnal variation of the $ABL_{CR}$ shows somewhat similar pattern as the surface temperature (Fig. 9c) and the LCL (Fig. 9d). Both surface temperature and LCL become maximum at 1400 IST and minimum at 0200 IST. The diurnal pattern of the LCL height is similar to the pattern of the surface temperature. The mean LCL height

is lower than the CBL and RL height. The higher the surface temperature higher is the LCL and vice versa.

Figure 10 shows the diurnal variation of the ABL height (obtained using $T$ variable) and surface temperature during different seasons. We have considered those cases which show the diurnal pattern of the ABL i.e. the formation of the CBL during daytime and SBL during nighttime as well as all those cases whenever SBL forms intermittently (excluding RL). As mentioned earlier, note that the occurrence frequency of the SBL just after the sunset is less than that of later periods at night. It means that the SBL has not formed always immediately after the sunset. Several times, formation of the SBL delay by 3-4 hours after the sunset. Thus, we cannot expect a perfect diurnal variation in all the cases, especially when the SBL formation is delayed. In order to study the diurnal variation of the ABL, we have segregated all the CBL and SBL observed at 3-h intervals during the diurnal cycle in different seasons. Similarly, all the cases of the CBL and RL observed at 3-h intervals are averaged into different seasons.

Figures 10a-10b show the diurnal variation of the $ABL_{CS}$ and $ABL_{CR}$ during different seasons. In general, the CBL heights vary between $1.05\pm0.46$ km to $2.25\pm0.72$ km, the SBL heights vary between $0.15\pm0.05$km to $0.33\pm0.27$kmand the RL heights varyfrom$1.39\pm0.41$ km to $1.94\pm0.63$ km. We have also obtained the diurnal variation of the surface temperature during different seasons as shown in Fig.10c. In general, the diurnal patterns of the $ABL_{CS}$ and $ABL_{CR}$ during different seasons are same as annual mean pattern shown in Fig 9a and Fig 9b, respectively. The diurnal variation of the $ABL_{CS}$ shows a seasonal pattern such that CBL attain maximum height during the summer monsoon while the SBL during the winter to pre-monsoon. The amplitude of the diurnal evolution of the $ABL_{CS}$ is stronger during pre-monsoon when compared to other seasons, i.e. maximum to minimum height variation is more during pre-monsoon when compared to other seasons. The SBL attains maximum height at 0200 IST. Fig 10b shows that the $ABL_{CR}$ height has a weak diurnal pattern during the winter when compared to summer. It is interesting to note that the CBL and RL heights reverse their seasonal pattern. The CBL height is higher during the summer and is lower during the winter in contrast, to the RL height, which becomes higher during the winter and lower during the summer. In general, the CBL attains maximum height at 1400-1700 IST and minimum during 0800-1100 IST during all seasons.

Figure 10c and 10d show the diurnal variation of the surface temperature and LCL during different seasons, respectively. Both the surface temperature and the LCL are the highest and lowest during pre-monsoon and winter, respectively. The highest amplitude of the diurnal variation of the $ABL_{CR}$ can be attributed to the highest surface temperature during pre-monsoon. Similarly, the weak diurnal pattern of the $ABL_{CR}$ can be attributed to the lowest surface temperature during the winter. In general, the LCL is lower than $ABL_{CR}$ throughout the year. The difference between the LCL and CBL height is more during the winter and post-monsoon than pre-monsoon and summer monsoon.

As mentioned earlier, the diurnal evolution of the annual and seasonal mean pattern of the ABL is closely associated with the surface temperature. In order to see their 3-hourly relationships, we obtained the scatter plot of the CBL, RL and SBL with the surface temperature as shown in Figs. 11a-c, respectively. Broadly, the scatter diagram indicates that warmer is the surface, the higher is the CBL and RL and vice versa (Figs. 11a-b). However, these features are not always consistent and several times they occur randomly. In contrast to the CBL and RL, SBL higher is higher over the colder surface and vice versa, however, these features also are not always consistent and several times they occur randomly (Fig. 11c). The corresponding 3-hourly relationship between the LCL and CBL, RL and SBL are shown in Figs 11d-c, respectively. The scatter plot between the CBL and LCL indicate that they occur randomly. The LCL generally occurs either below or at the CBL and RL except a

few times when it occurs above the CBL and RL (Figs. 11d-e). The cases when the LCL occurs above the CBL or RL, clouds may not be generated by the processes driven by the ABL and can be formed due to large scale-dynamics (Anurose et al., 2016). We observed no relationship between the SBL and LCL (Fig. 11f). For the SBL case, as the vertical motion is inhibited, the relationship between the LCL and SBL is irrelevant (Anurose et al., 2016). Anurose et al., (2016) also studied the relationship between the CBL height and the LCL over the coastal station, Thiruvananthapuram (8.5° N, 76.9° E), they did not observed any relationship. However, the LCL over Thiruvananthapuram is found to higher than ABL for a majority of the database in contrast to Gadanki.

Figures 12a and 12b show the schematic representation of the diurnal evolution of the mean ABL from 11:00 IST on the first day to 11:00 IST on the second day during the winter and summer monsoon seasons, respectively. The diagram is generated from the seasonal mean ABL height data presented in Figs.10a and 10b. The vertical cross section of the ABL characterizing the seasonal mean CBL, SBL, RL, and entrainment zone (E.Z.), capping inversion and the LCL obtained from the observed data over Gadanki. The schematic diagram represents the typical evolution of the boundary layer, consistent to the diagram presented in the Stull (1988) and Wallace and Hobbs (2006). The CBL during the winter evolves slowly when compared to summer monsoon season in which the ABL growth is rapid. The SBL starts to form well before the sunset during both seasons; however, it remains persistent, even after the sunrise only during the winter season. During the winter, the RL remains almost constant throughout the night. However, during the summer, the RL rapidly decreases as the night passes. The capping inversion during the summer is thicker when compared to the summer monsoon. We have also shown the seasonal mean LCL which occurs within the CBL and RL during both seasons. Note that the transition regions (from the CBL to RL during evening transition and the RL to CBL during morning transition) cannot be accurately represented with available time resolutions. Thus, the part of the CBL after the sunset and the part of the RL after the sunrise may not possess any meaning. This schematic diagram clearly represents the typical (Fig.4), annual mean (Fig.9) and seasonal mean (Fig.10) characteristic of the ABL.

**3.7 Qualitative relationships between cloud top height (CTH) and $ABL_{CR}$ height**

The presence of the clouds has a large impact on the boundary layer structure. However, it leads to considerable complication because of the important role played by radiative fluxes and phase change (Garratt, 1992). The relationship between the CTH and the CBL/RL is obtained and is shown in Fig. 13. We have observed the CTH at various layers ranging from within the ABL to up to 12 km during the different campaigns. The CTH relative to the $ABL_{CR}$ is obtained for each individual campaign, which is listed in Table1. In total, for 630 cases the clouds were present either below or near to or above the $ABL_{CR}$. Rest of 50 cases are when the clear sky conditions observed. Note that in total 680 cases, when CBL and RL are defined. We observed that in total 175, 199, 222 and 34 cases, when CTH occurred within ± 0.5 km, below 0.5 km, above 0.5 km but below 6.0 km, and above 6.0 km of the $ABL_{CR}$, respectively. These cut off heights regions are selected through visual inspection by trial and error after examining several ABL height and CTH timeseries. The timeseries of the CTH, ,$ABL_{CR}$ and LCL for the cases when the CTH occurred within ± 0.5 km, below 0.5 km, above 0.5 km but below 6.0 km of $ABL_{CR}$ are shown in Fig 13. The CTH within ± 0.5 km of the $ABL_{CR}$ is positively correlated (r = 0.88) with SD of 0.28 km (Fig 13a). Fig 13a further reveals a clear association between the $ABL_{CR}$ and the CTH variations. These cases indicate the cloud topped boundary layer (CTBL) where clouds are limited in their

vertical extent by main capping or subsidence inversion (Garratt, 1992). Fig. 13b shows that the CTH occurring below 0.5 km of the $ABL_{CR}$ is also positively correlated (r = 0.71) with SD of 0.37 km. Although these clouds occur well below the $ABL_{CR}$, both vary in the similar fashion (Fig 13b). It can be seen that higher the cloud level higher the $ABL_{CR}$ and vice versa. Similarly, the $ABL_{CR}$ height variation is also well correlated (r = 0.58) with the CTH variation occurring above 0.5 km but below 6.0 km of the $ABL_{CR}$ (Fig 13c).

When the CTH is within ± 0.5 km of the $ABL_{CR}$, LCL occurs mostly below the ABL, except a few cases when it coincides with either the ABL or CTH (Fig.13a). When the CTH is below 0.5 km of the $ABL_{CR,}$ the LCL again occurs mostly below the ABL but generally coincided with the CTH (Fig.13b). In this case, the LCL sometimes also occurs above the CTH. The clouds occurring below the ABL could be the shallow clouds, in such cases LCL representing the cloud base may occur near to the CTH. However, it is to be noted that the

CTH represents the cloud condition for the area averaged over $0.25^{o}$ latitude X $0.25^{o}$ longitude regions, whereas the LCL indicates the cloud base exactly over the observation site. Thus, the LCL may not always agree with the CTH when the cloud is not extended over the larger area. For the cases when the clouds occurring above 0.5 km but below 6.0 km of the $ABL_{CR}$, the LCL mostly occurs either below the ABL or generally coincides with the ABL (Fig 13c).

It is interesting to note that when the CTH is below the $ABL_{CR}$, CBL and RL occur at higher height (mostly above 2 km) whereas when the CTH is above the $ABL_{CR}$, CBL and RL occur at a lower height (mostly below 2.0 km). Generally, the CBL (sometimes also called as fair weather boundary layer) occurs at lower height during a shallow cumulus when compared to clear sky conditions (Medeiros et al., 2005). Very deep convective clouds do not show any relationship with the $ABL_{CR}$ variation (figure not shown) as can be seen from Fig. 4b.

Qualitatively, it indicates that the presence of the clouds near to the CBL and RL directly impact its variation, but not the high level clouds due to the deep convection events. It is to be noted that there are two occasions when rainfall occurred during the campaign periods, however, these few data do not reveal any relation between the ABL and rainfall.

**4 Discussion and Conclusions**

The unique and long term intensive campaigns of high vertical resolution radiosonde observations on multiple of 3 hours over a tropical location, Gadanki, in the Indian monsoon region reveal the clear diurnal structure of the ABL height. The high vertical resolution of the radiosonde data enables us to detect the SBL height directly, which otherwise was very difficult.

     Identification of the ABL is generally preferred using $\theta_v$ and $q$ obtained from radiosonde observation because

they can represent the mixing height better than the $T$. However, we observed an excellent correlation between $T$ and $\theta_v$ suggesting that ABL can be identified using $T$. Moreover, use of $T$ can give both elevated inversion as well as surface based inversion well suited to study the diurnal variation of the ABL. The limitation of using $T$ is that it can also identify mid-level inversions sometimes. However, to avoid mid-level inversions, if any, we have restricted the ABL height identification below 3.5 km. In case of multiple inversions, the lower one having 80% of main inversion is considered. The correlation between $T$ (or $\theta_v$) and $N$ are in good agreement with Basha and

Ratnam, (2009). We also found that $N$ yields the ABL height lower than that of $T$ several times, but not always, in contrast to Chennai located 120 km southeast of Gadanki where significant ABL height difference of ~ 0.84 (between $\theta$ and $N$) is observed in evening soundings (Seidel et al., 2010).

The nighttime ABL is complex to define when compared to daytime ABL. As the nighttime ABL or the SBL depends on the surface cooling, if it delays, or does not form at all, which generally happens, one will find it as about same as pervious daytime ABL (i.e CBL). It is the part of the daytime mean state and has not formed due to action of nightime surface forcing. However, as it is above the surface, which is an only criterion left to assign the RL as the ABL in the absence of the SBL(Liu and Liang, 2010). But, if the measuring instrument has limited capability to detect the SBL, one will land in defining the RL as nighttime ABL, which will not be at rue representation of the ABL height.

In total, the SBL forms about 50% times during the midnight to morning. In the early part of the night, the SBL occurs less frequently (33%) than late night and hence indicates the delay in the surface cooling process. The SBL forms more frequently during winter season when compared to other seasons. Thus, diurnal variation of the ABL occurs more often during winter than the summer. There could be various reasons for the delay in the SBL formation, such as cloud cover or wet surface due to rain which can disturb or delay the surface cooling process. Sandeep et al, (2015) observed that the ABL over Gadanki after the sunset becomes shallower and its growth delayed by 1–4 h during wet episodes. Over Indian region, clouds and precipitation most frequently occur during the summer monsoon season when compared to the winter season, suggesting the formation of the SBL will be less frequent during the former season consistent to the observed result. However, irrespective of the season the surface temperature shows a diurnal pattern. Thus, another possibility of less occurrence of the SBL during summer monsoon season could be due to high night time temperature. In fact the nighttime surface temperature during summer monsoon as well as post and pre monsoons are greater than that of daytime surface temperature during the winter season. Though, the surface temperature during these seasons decreases during nighttime, it doesn't have sufficient cooling effect as winter season probably preventing the formation of the SBL most of the time during the summer season. Thus, it could be possible that even though the surface temperatures show diurnal variation during the summer monsoon, diurnal variability in the ABL may not be expected.

The minimum height of the CBL at 0800 IST is due to weak convection (thermals) during the morning hours when compared to other timings of the day. As convection becomes stronger (the strongest surface warming at 1400 IST) due to the strong thermals, the CBL becomes higher and reaches a maximum height at 1400-1700 IST. Similarly, as the night passes, surface cooling becomes stronger (the strongest cooling occurs at 0200 IST) leading to higher SBL height at 0200 IST -0500 IST. The RL remains sometimes at similar height as the CBL in case of very strong gradient in moisture and temperature. But generally it lowers down as time passes during the night, especially during the summer monsoon season. As daytime convection is switched off during the night, the turbulence strength goes weaker and weaker as night passes leading to decrease in the RL height.

During the pre-monsoon, the surface temperature has the strongest diurnal variation which manifests stronger diurnal variation of the $ABL_{CR}$ height (Angevine et al., 2001). But higher CBL occurs during the summer monsoon season and not during the pre-monsoon. The higher CBL during summer monsoon season is due to stronger convection occurring in this season when compared to the other seasons. During the winter season, the surface temperature is low leading to a weak diurnal pattern of the $ABL_{CR}$ (Angevine et al., 2001). Since convection is weaker during the winter season, the CBL is at a lower height when compared to the other seasons. The reversal of the CBL and RL height patterns between summer monsoon and winter is due to the surface temperature variation and strength of convection. It is due to the fact that during the winter the RL does

not lower as observed for the other seasons due to less surface temperatures, while the CBL becomes higher during summer monsoon seasons due to stronger convection when compare to the winter. Thus, the CBL is lower during the winter, but it is higher during summer monsoon while RL is higher during the winter but it is lower during the summer monsoon season.

Finally, the qualitative relationship between the $ABL_{CR}$ height, the CTH and the LCL is examined. We only provide here qualitative information based on the CTH obtained using merged TBB data, since direct observation of the CTH over the launch site during the campaign periods are not available. As observed in this study, the CTH at various layers has been observed using ceilometer at Ahmedabad (23.03∘ N, 72.54∘ E), India (Sharma et al., 2016). As suggested by Wang and Rassow (1995), the various cloud layers can be obtained utilizing the RH data (Wang and Rassow, 1995). However, as the criteria for fixing the cloud base and top heights using the RH data has not been finalized for the tropical clouds occurring over this region, we have preferred satellite derived CTH data in this study. We also obtained the LCL height, but for majority of the database LCL occurs below the ABL and they are randomly related consistent to Anurose et al., (2016). Thus, we have interpreted our results based on the CTH data only. If the clouds occur above the $ABL_{CR}$ during the daytime, they will absorb the incoming solar radiation and hence cool the surface. This in turn will weaken the thermals and hence decrease the CBL height. On the other hand, when the clouds occur below the CBL, it will cool the surface, but warm the region between cloud top and CBL and hence can strengthen the thermals which will lead to increase the CBL height. This explains why CBL is at lower height when the clouds above it and at the higher height when clouds below it. If the cloud occurs during nighttime, the situation will be more complex and difficult to explain the RL variability. During nighttime, the clouds will block the outgoing long wave radiation, which in general warm below the RL and hence disturb the surface cooling and the formation of the SBL. The verification of the CTH height using space borne satellites and ground based observations such as ceilometer over the launch site will be carried out as a separate study in the future.

Following are the main findings on the diurnal variability of the atmospheric boundary layer (ABL):

1. The convective boundary layer (CBL) height has a large variation ranging from as low as 0.4 km to as high as about 3.0 km above the surface and occurs uniformly at 3-h intervals during the diurnal cycle over Gadanki, a tropical station in the Indian monsoon region.

2. The stable boundary layer (SBL) mainly forms during night time; however, it can also form during daytime, especially during evening and morning hours, i.e. during transition periods. The SBL forms about 50% of times of total observations during 2300-0500 IST. At 2000 IST, occurrence of the SBL is only 33%, indicating that delay in surface cooling for about 17% of the times. About 25 % of the time the SBL forms at 0800 IST indicating the dominance of the surface cooling even after the sunrise. As surface cooling starts well before the sunset, sometimes the SBL (about 9%) also forms at 1700 IST.

3. The overall mean SBL lies well within the 0.3 km and the mean CBL lies well within 3.0 km consistent with the available literature. However, the maximum probability distribution of the SBL occurs at 0.15 km lower than its mean value. In contrast to the SBL, the maximum probability distribution of the CBL coincides with mean CBL at about 2 km for the winter season and the whole year. The maximum

probability distribution of the CBL during the summer monsoon season has a broader peak when compare to winter season.

4. The CBL and the RL heights obtained using different methods ($T$, $\theta_v$, $q$ and $N$) correlates well.

    5. A clear diurnal variation of the $ABL_{CS}$ height over the different seasons is observed with the maximum CBL height during the summer monsoon season while the maximum SBL height during the winter to pre-monsoon. The seasonal pattern reverses for the RL height that becomes higher during the winter and lower during the summer monsoon season.

6. The $ABL_{CR}$ height is positively correlated with the CTH occurring near to it, however, the deep convective clouds do not show any relationship. When the cloud is at lower height the $ABL_{CR}$ is relatively higher and vice versa. This needs to be verified using independent observations of the CTH perhaps using Ceilometer observations.

    7. Over Gadanki, the LCL occurs below the CBL and RL for the majority of the database and they are
randomly related.

*Acknowledgements.* We thank NARL radiosonde operation team for conducting the experiment under tropical tropopause dynamics (TTD) campaigns fully supported by the Indian Space Research Organization as a part of CAWSES India Phase-II program.

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

**Table 1: List of 3-days campaigns conducted during the period December 2010-March 2014. Total number of observations, rejection of bad quality data, the SBL, the CBL and the RL are listed for each campaign. Shaded regions with italic fonts indicate campaigns with more than 90% of the SBL defined. Bold indicates those campaigns where SBL is not defined and the rest when the SBL occur intermittently. CTH relative to ABL are also listed.**


| SN | Period | No of Observations | | | | |
|---|---|---|---|---|---|---|
| | | Total | Rejected | SBL | CBL | RL | CTH-ABL |
| 1 | *28 Dec 2010-31 Dec 2010* | *24* | *0* | *10* | *12* | *12* | *±1.5* |
| 2 | *17 Jan 2011-20 Jan 2011* | *24* | *0* | *14* | *10* | *6* | *-2.0-1.0* |
| 3 | *08 Feb 2011-11 Feb 2011* | *24* | *0* | *14* | *11* | *7* | *-2.0-1.0* |
| 4 | **26 Apr 2011-29 Apr 2011** | **24** | **2** | **0** | **11** | **10** | **-1.0-2.0** |
| 5 | 18 May 2011-21 May 2011 | 24 | 1 | 5 | 8 | 9 | -1.0-7.0 |
| 6 | **20 Jun 2011-23 Jun2011** | **24** | **0** | **0** | **12** | **12** | **-1.5-1.0** |
| 7 | **21 Jul 2011-24 Jul 2011** | **24** | **0** | **0** | **12** | **12** | **-1.5-2.0** |
| 8 | 17 Aug 2011-20 Aug2011 | 24 | 1 | 5 | 10 | 9 | -1.5-1.0 |
| 9 | *12 Sep 2011-15 Sep 2011* | *24* | *1* | *11* | *12* | *8* | *±2.0* |
| 10 | 12 Oct 2011-15 Oct 2011 | 12 | 2 | 6 | 5 | 3 | ±0.5 |
| 11 | *14 Nov 2011-17 Nov 2011* | *24* | *0* | *13* | *12* | *12* | *-2.0-0.0* |
| 12 | 08 Dec 2011-11 Dec 2011 | 23 | 1 | 7 | 11 | 11 | -2.0-8.0 |
| 13 | 18 Jan 2012-21 Jan 2012 | 17 | 1 | 8 | 7 | 9 | ±2.0 |
| 14 | *23 Feb 2012-26 Feb 2012* | *22* | *0* | *13* | *10* | *11* | *-1.5-2.0* |
| 15 | **29 Mar 2012-01 Apr 2012** | **23** | **0** | **0** | **12** | **11** | **-1.0-4.0** |
| 16 | 01 Apr 2012-04 Apr 2012 | 24 | 0 | 9 | 11 | 7 | 0.5-2.5 |
| 17 | 29 May 2012-01 Jun 2012 | 23 | 0 | 6 | 12 | 11 | -0.5-1.5 |
| 18 | 26 Jun 2012-29 Jun 2012 | 23 | 0 | 8 | 10 | 8 | -1.0-2.5 |
| 19 | 24 Jul 2012-27 Jul2012 | 24 | 2 | 4 | 11 | 11 | -1.5-1.0 |
| 20 | 21 Aug 2012-24 Aug 2012 | 23 | 0 | 8 | 10 | 6 | -1.5-2.5 |
| 21 | 12 Sep 2012-15 Sep 2012 | 23 | 0 | 7 | 11 | 8 | -0.5-2.0 |
| 22 | 03 Oct 2012-06 Oct 2012 | 21 | 0 | 6 | 11 | 10 | ±0.5 |
| 23 | 05 Nov 2012-08 Nov2012 | 12 | 1 | 2 | 6 | 5 | ±0.5 |
| 24 | 21 May 2013-24 May 2013 | 22 | 1 | 6 | 10 | 10 | -1.0-2.5 |
| 25 | 17 Jun 2013-20 Jun 2013 | 24 | 0 | 2 | 12 | 12 | -1.0-3.5 |
| 26 | 15 Jul 2013-18 Jul 2013 | 24 | 0 | 5 | 12 | 12 | -1.0-2.0 |
| 27 | 26 Aug 2013-29 Aug 2013 | 24 | 0 | 5 | 11 | 7 | -1.5-0.5 |
| 28 | 23 Sep 2013-26 Sep 2013 | 24 | 0 | 4 | 11 | 6 | ±1.5 |
| 29 | 28 Oct 2013-31 Oct 2013 | 21 | 0 | 6 | 11 | 9 | -2.0-5.0 |
| 30 | **21 Nov 2013-24 Nov 2013** | **19** | **2** | **0** | **8** | **9** | -2.0-5.0 |
| 31 | *18 Dec 2013-21 Dec 2013* | *24* | *0* | *11* | *12* | *12* | *1.0-11.0* |
| 32 | **27 Jan 2014-30 Jan 2014** | **24** | **0** | **0** | **12** | **12** | **-0.5-(-2.0)** |
| 33 | **24 Feb 2014-27 Feb 2014** | **24** | **0** | **0** | **12** | **12** | **-2.0-0.5** |
| 34 | *25 Mar 2014-28 Mar 2014* | *24* | *1* | *12* | *12* | *11* | *-1.0-7.0* |
| | Total | 764 | 17 | 207 | 360 | 320 | |

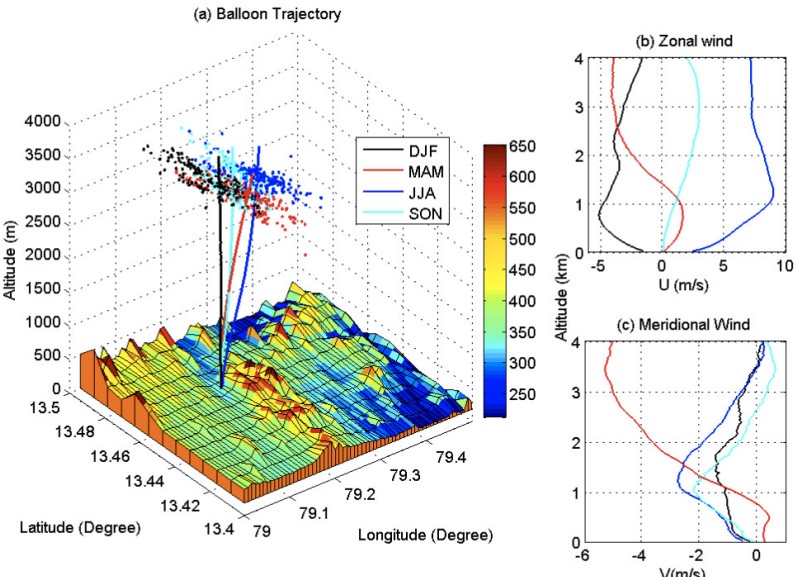


**Figure 1: (a) Topographical features surrounding the location of radiosonde launch site, Gadanki (13.45$^o$ N, 79.2$^o$ E), a tropical station, India, along with profiles of seasonal mean trajectories (thick lines) from the surface to 4 km and the locations of the balloon reaching at altitude 4 km (dots) during different seasons. The mean (b) zonal wind and (c) meridional winds for different seasons observed during the period December 2010-March 2014. Global 30-arc-second gridded topography data is provided by the National Centers for Environmental Information, USA.**



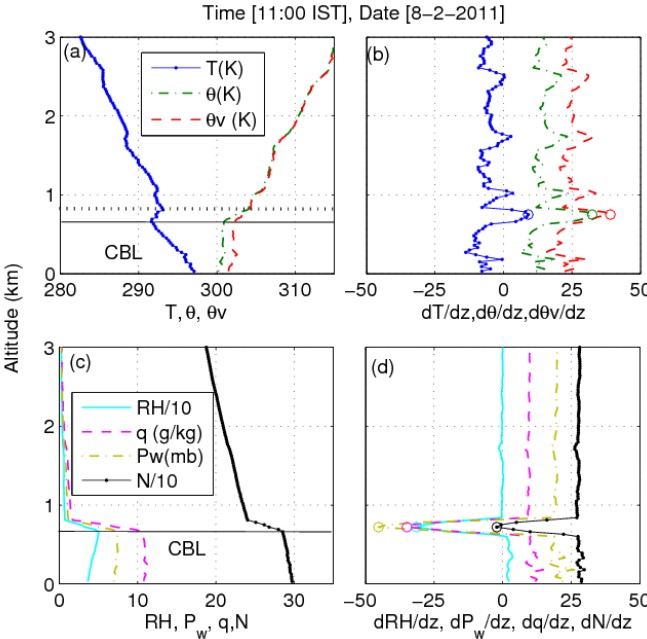

**Figure 2:Typical profiles of (a) temperature (*T*), potential temperature (*θ*) and virtual potential temperature (*θ$_v$*) and (c) relative humidity (RH), specific humidity (*q*),  water vapor pressure (*P$_w$*), and radio refractivity (*N*) showing convective boundary layer (CBL) using GPS radiosonde observation at 1100 IST on  February 8, 2011 over Gadanki (13.45ºN, 79.2ºE). (b) and (d) show the gradient profiles corresponding to (a) and (c), respectively. Solid horizontal lines in (a) and (c) indicates the base of the inversion layer. Dotted line in (a) indicates the top of the inversion. Open circles in figures (b) and (d) denote the CBL heights above ground level.**


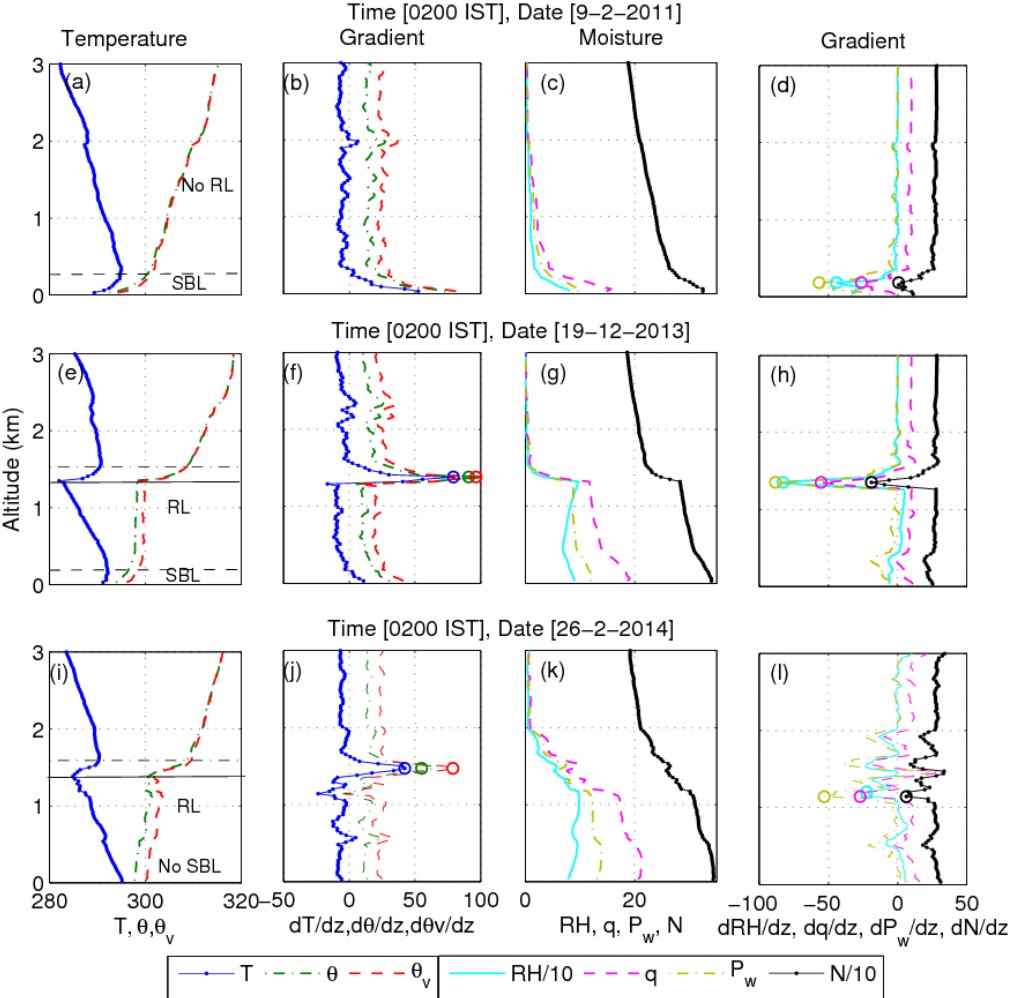


**Figure 3: Same as Figure (2) but for night time profiles observed at 0200 IST indicating (a) the stable boundary layer (SBL) only, but not the residual layer (RL) observed on February 9, 2011, (b) both the SBL and the RL observed on December 19, 2013 and (c)  the RL but not the SBL observed on February 26, 2014. The horizontal dashed lines denote the surface based inversion in the temperature profile and open circles denote maximum gradients in temperature variables or minimum gradients in the moisture variables. Horizontal dash-dotted lines indicate capping inversion layer.**


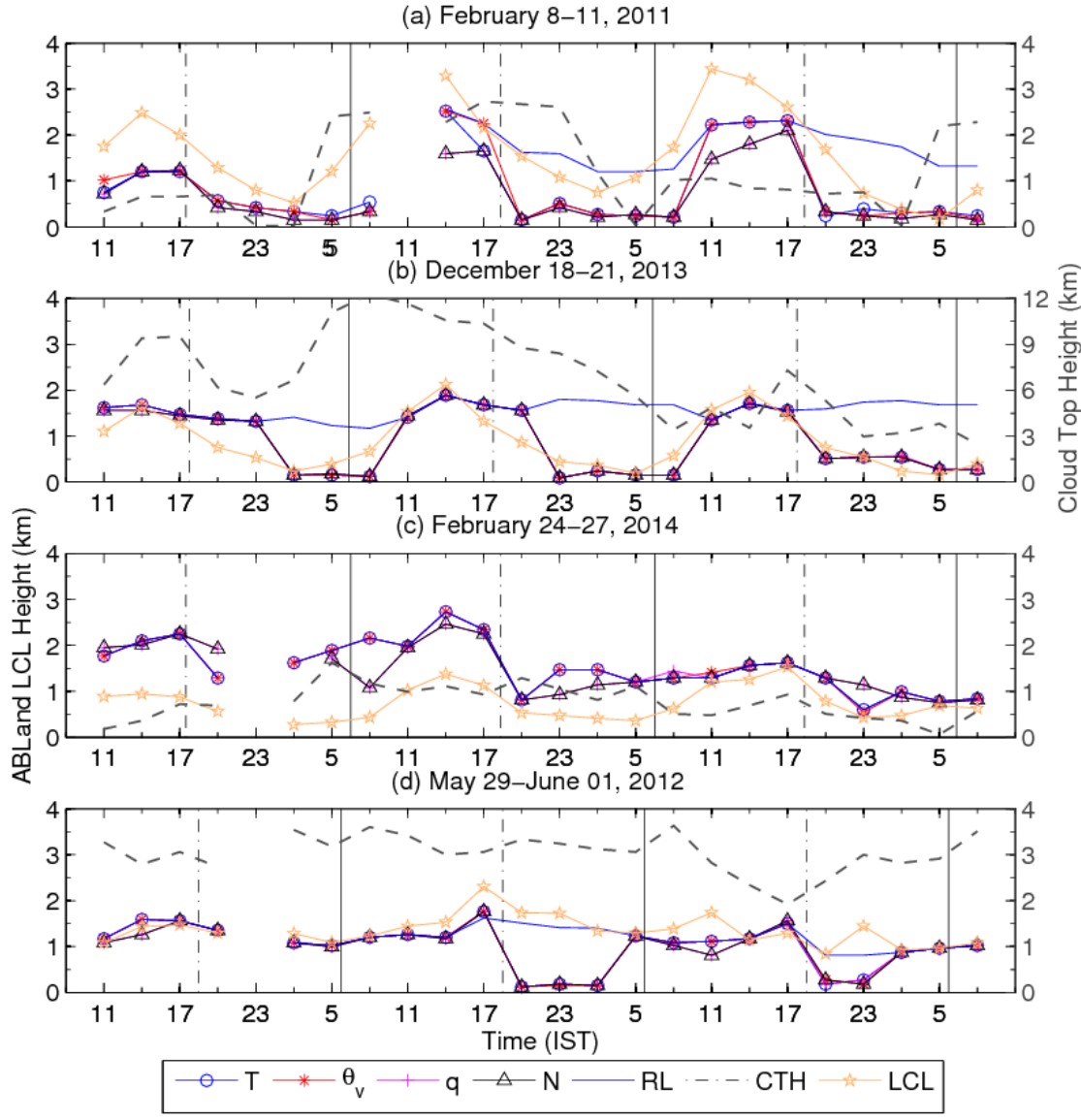

**Figure 4:** Diurnal variability of the ABL (CBL, SBL and RL) height obtained from the variables *T*, *θ<sub>v</sub>* *q*, and *N* during (a) February 8-11, 2011, (b) December 18-21, 2013, (c) February 18-21, 2013, and (d) May 29- June 01, 2012. Thick dashed lines indicate the cloud top height (CTH). Dark yellow star line indicates the lifting condensation level (LCL). Vertical dashed and solid bars indicate sunset and sunrise times, respectively.


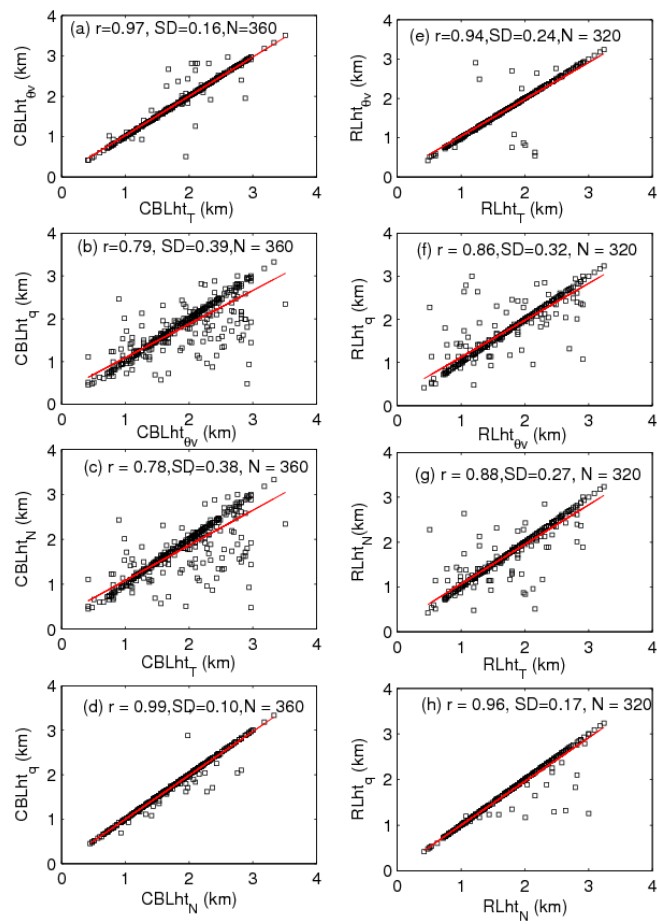

**Figure 5: (Left panels) Correlation between the CBL heights obtained using (a)** *T* **and** *θ*<sub>*v*</sub>**,(b)** *θ*<sub>*v*</sub>**and** *q*, **(c)** *T* **and** *N,* **and(d)** *N* **and** *q,* **(Right Panels) (e)-(h) are same as (a)-(d) but the RL heights.**

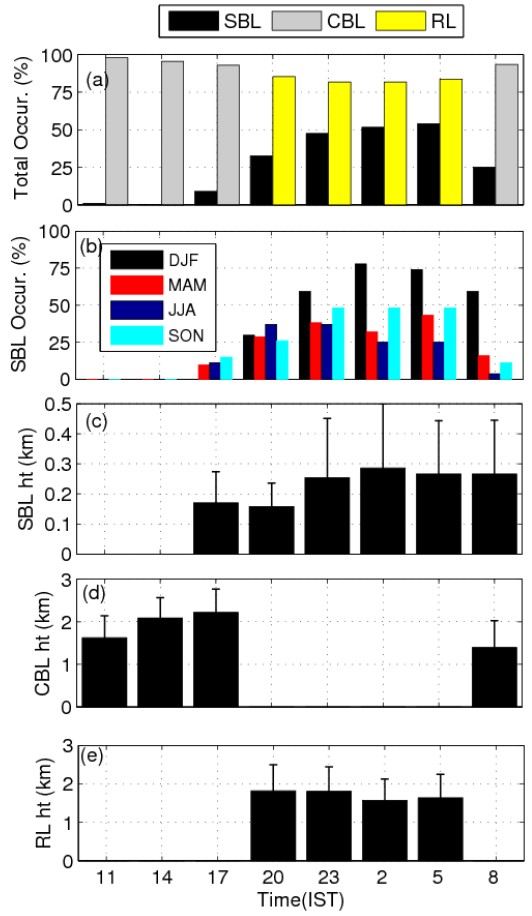


**Figure 6: (a) Total percentage occurrence of the SBL, CBL, and RL heights (b) percentage occurrence of the SBL during different seasons. Mean and standard deviations of (c) the SBL height observed at 3 hours interval between 1700 IST and 0800 IST, (d) the CBL observed at 3 hours interval between 0800 IST and 1700 IST and (e) the RL observed at 3 hours interval between 2000 IST and 2300 IST. These statistics are obtained using $T$ variables.**


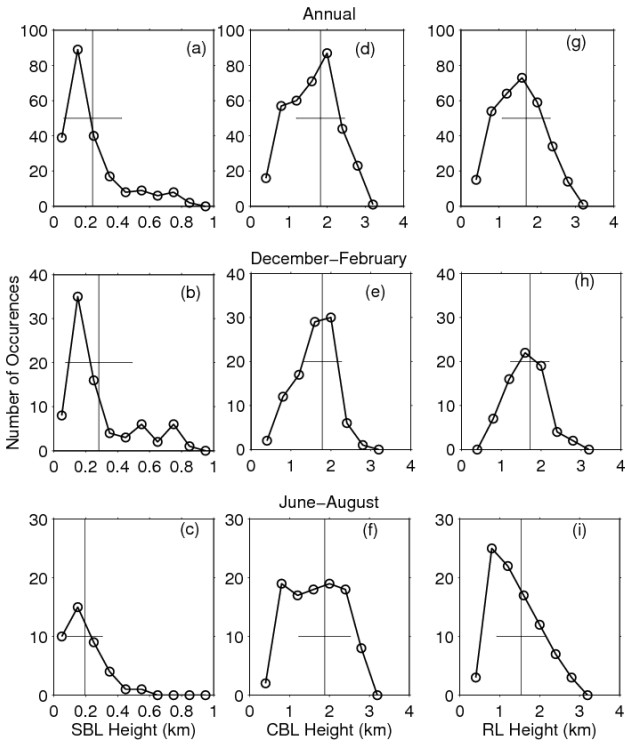

**Figure 7: Probability distribution of 3 hourly (left column panels) SBL height with 0.1 km interval obtained for (a) annual (b) winter and (c) summer from the data observed during December 2010-March 2014. (d)-(e) same as (a)-(c) but for the CBL height, and (f)-(h) are same as (a)-(c) but for the RL height with 0.4 km interval. The Points at 835 which the vertical line intersects the x-axis represents their mean heights while the length of the horizontal bar represents the corresponding standard deviations.**





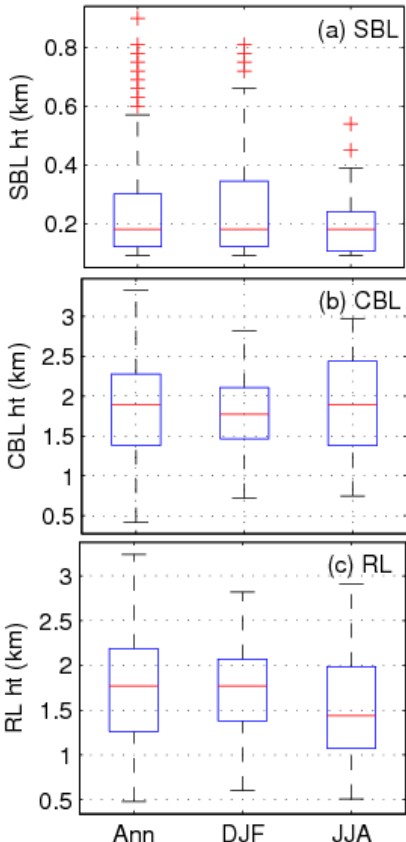

**Figure 8: Boxplot (representing median and interquartile ranges) of (a) SBL height (b) CBL height and (c) RL height**
**for annual (Ann) winter (DJF) and summer monsoon (JJA) from the data observed during December 2010-March 2014.**

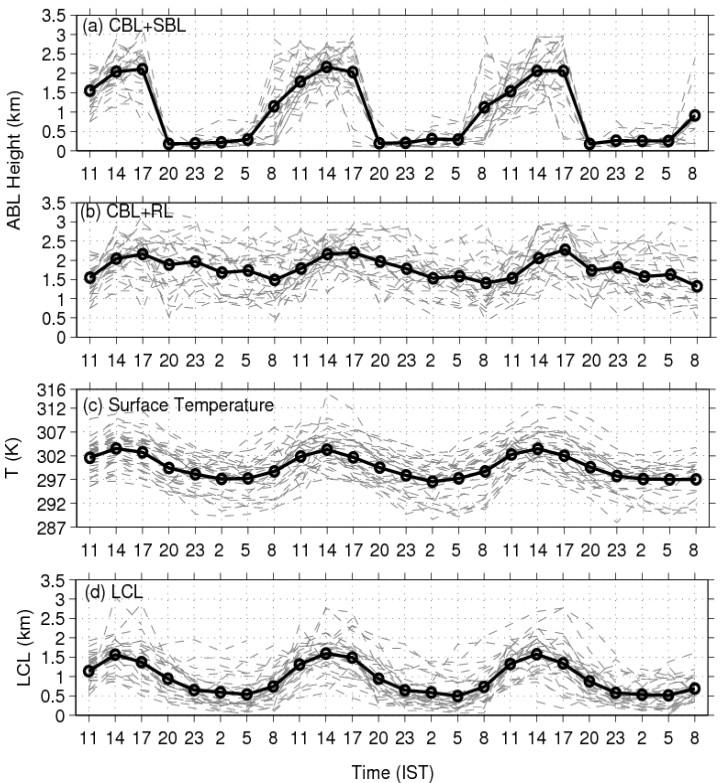

**Figure 9: (a) The 3 days diurnal variability of (a) the ABL (CBL+SBL) considering all the data observed for the CBL and SBL, (b) the CBL+RL considering all the data observed for the CBL and RL obtained using *T* variable, (c) the surface temperature, and (d) the LCL height observed during the period December 2010-March 2014.**

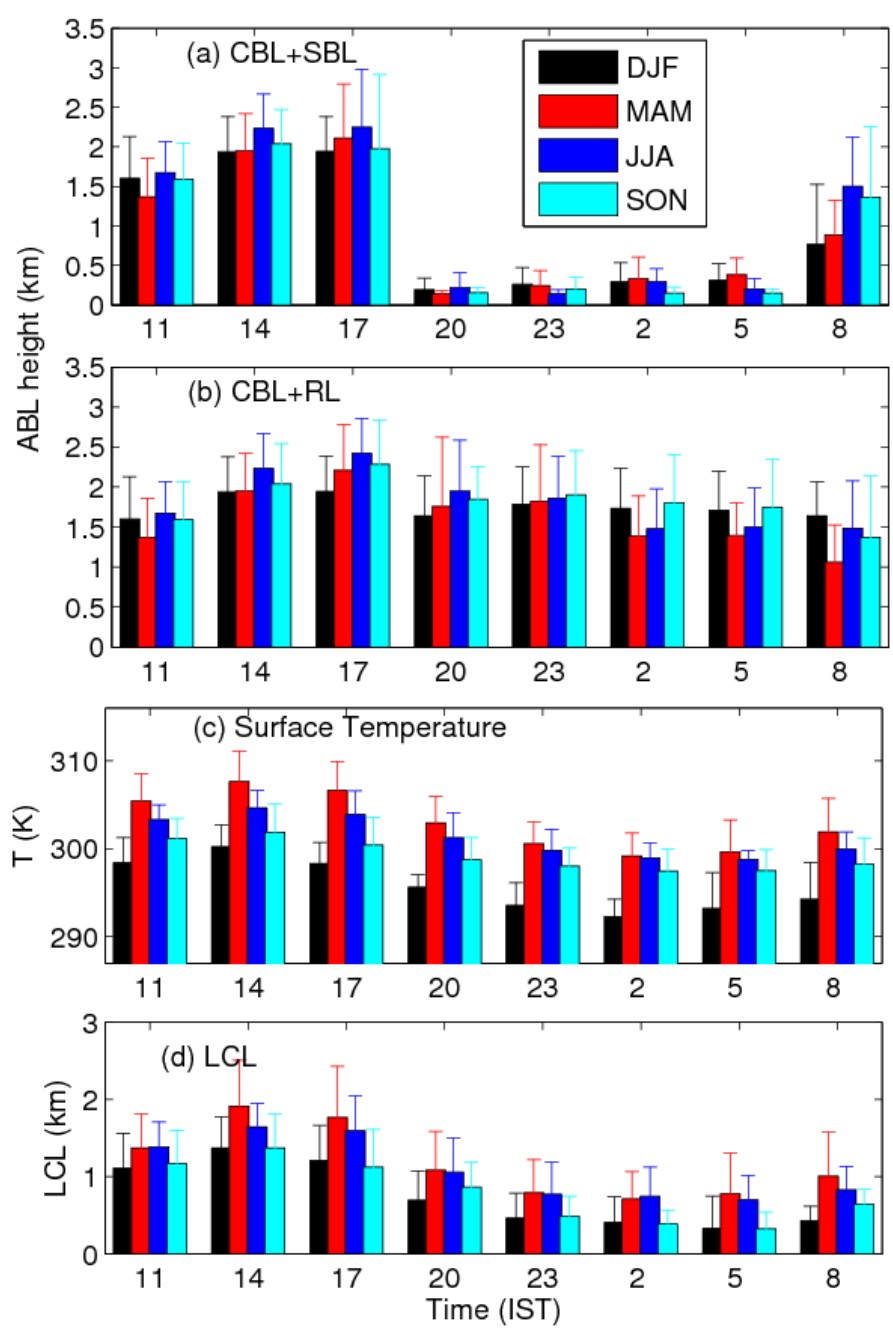

**Figure 10: The diurnal variation of (a) the ABL$_{CS}$ (CBL+SBL) height, (b) the ABL$_{CR}$(CBL+RL) height   (c) the surface temperature and (d) the LCL height during different seasons.**


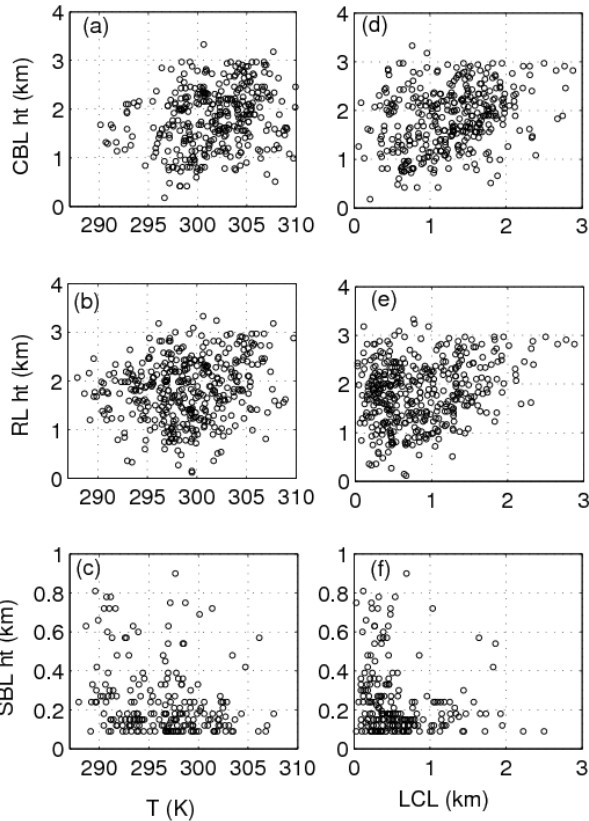

**Figure 11: Scatter plot between the surface temperature and (a) CBL, (b) RL, and (c) SBL. (d)-(f) are same as (a)-(c) but for the LCL.**


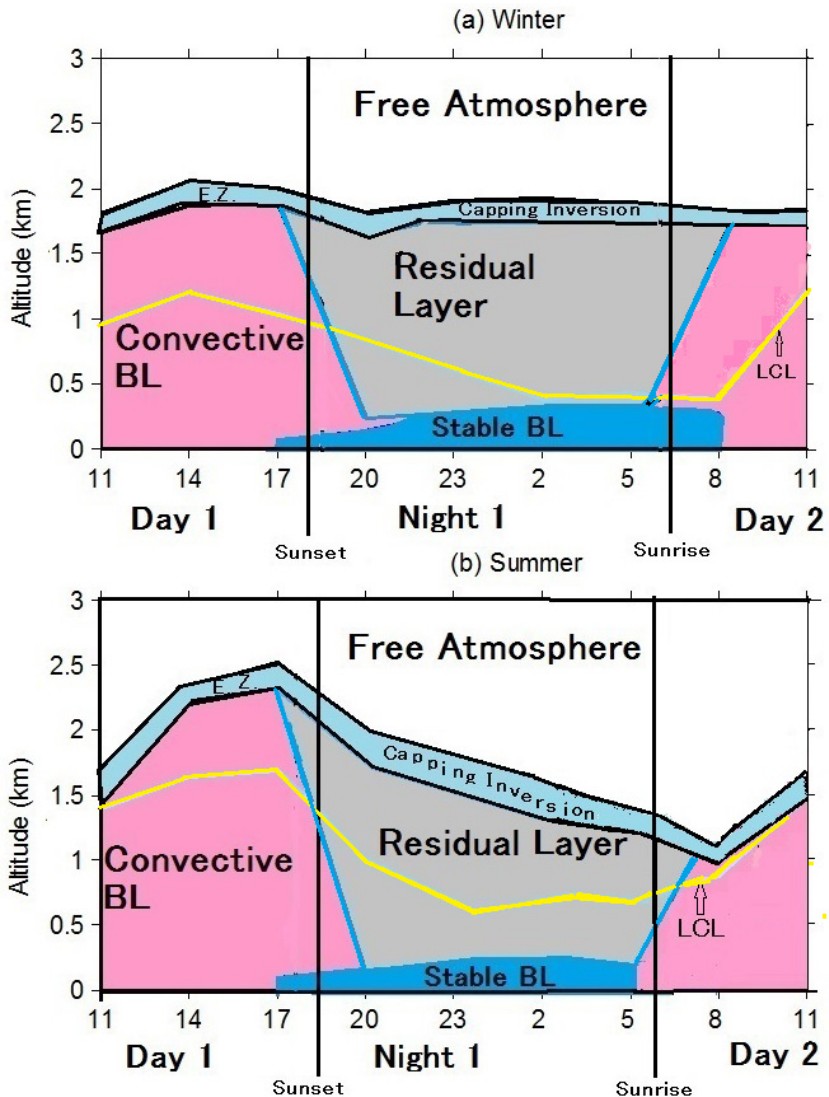

**Figure 12: Schematic diagram of the vertical cross section of the mean ABL structure during the (a) winter (DJF) and summer monsoon (JJA) seasons over Gadanki. E.Z. indicates entrainment zone.**

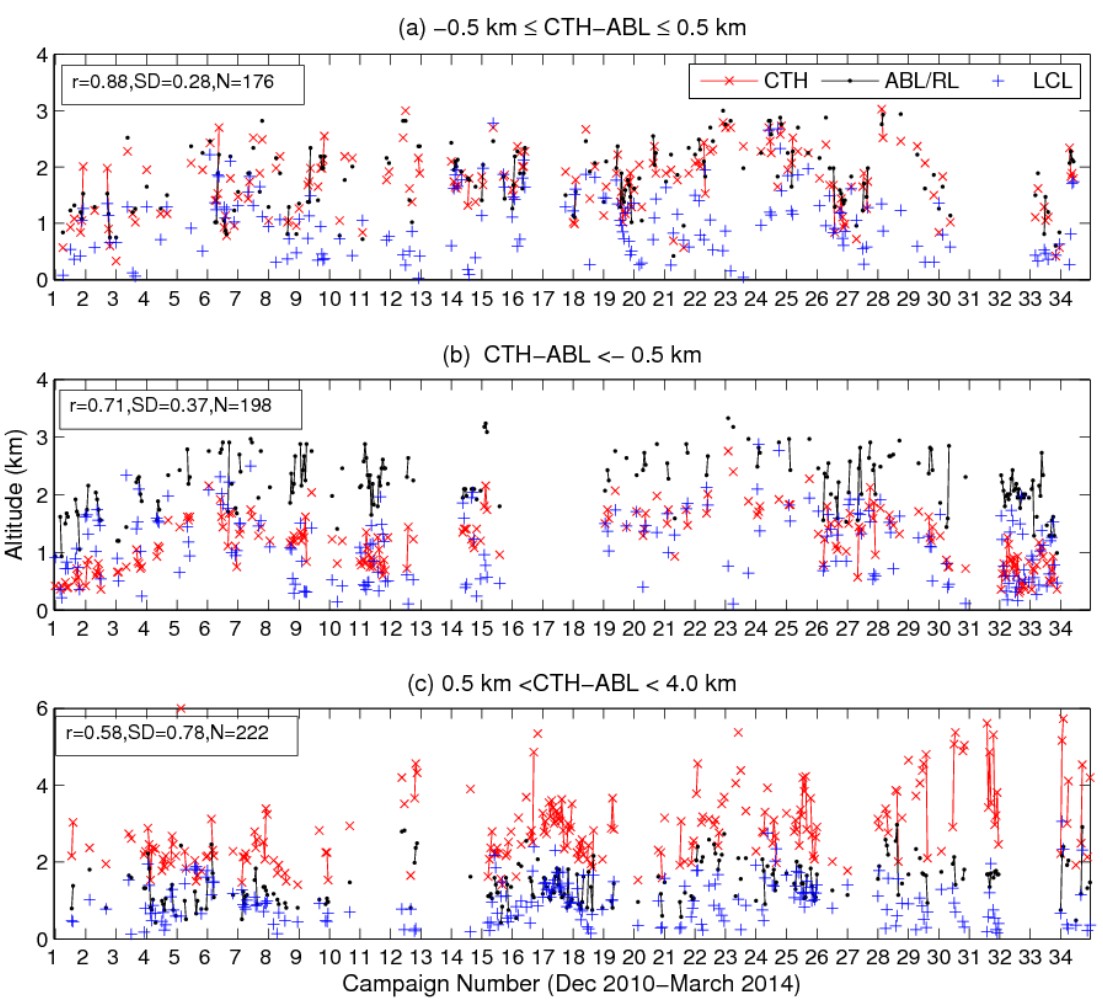

**Figure 13: Time series of the CTH, the ABL$_{CR}$ top and the LCL observed for the cases (a) -0.5 km ≤CTH-ABL≤0.5 km (b) CTH-ABL<-0.5 km and (c) 0.5 km <CTH-ABL<4.0 km.**
