# Peer review of "Diurnal variability of the Atmospheric Boundary Layer"

_Atmospheric Chemistry and Physics, 2016_

## Referee Comment (RC1) · Anonymous Referee #1 · 27 Jul 2016

This study presents an analysis of 3-hourly radiosonde data taken over 3-day intervals in each month over several years. The focus is on identifying the boundary layer height. This height is divided into convective and stable boundary layers and also residual layers. Several methods are used to identify the top of the ABL, but they are all based on gradients. These methods are appropriate, and the comparison of the methods is a nice aspect of this study. The diurnal cycle of ABL height is described; the upshot seems to be that there is a strong diurnal cycle but the amplitude is affected by the season as well as the presence of clouds, which also appear to affect the phase. Overall, the presentation of the results is clear, but the weakness is that it is not clear whether the results provide novel insight. Instead, the novelty appears to be in the data set

itself. There are several areas where some additional work could improve the analysis. These are relatively minor issues. The text is written well, but could stand another round of close editing for small grammatical and English issues (some examples listed below).

Perhaps the main issue I have with this study is that it is quite focused on the radiosondes, with limited support from other observations. This becomes crucial as the text explores the impact of clouds on the ABL structure. There is a good use of IR brightness temperature to provide an estimate of cloud top height, but this is the only cloud observations that are presented. I found that to be surprising. Perhaps even more surprising once I visited the NARL website (https://www.narl.gov.in/) and found that there are several instruments that could provide useful supplementary data. One that could provide highly complementary data is the microwave radiometer; the web site says that it retrieves cloud base and liquid water path. These could be quite useful for more clearly defining the cloud layer. There are also radiation sensors and eddy covariance latent and sensible heat fluxes that could be used to construct a surface energy balance. There are also rain gauges and a disdrometer, which could be used to explore the ABL height as a function of rain rate. Such an analysis could bolster the conclusions about deep convection having little impact on the ABL height.

The histograms of Figure 7 raise an issue about the statistics being used. Most of the histograms (which are NOT pdfs) look very non-Gaussian. The text mentions that the peak of the SBL histogram is at a substantially lower altitude than the mean (also true for RL in summer). Based on these histograms, I suggest also reporting the median and interquartile range, which provide a better estimate of the typical values and variability of the data.

I found the definition of the residual layer (RL) to be a little unclear. It seems to be defined exactly the same as for the CBL, is that correct? It would be good to include an explanation in Section 2.3.

One aspect of the residual layer that has been pointed out as being important for the diurnal evolution of the ABL is that it provides the potential for "explosive growth" of the ABL as a CBL forms in the morning and grows into the RL. This was not mentioned in this paper. Is it possible to quantify whether this explosive growth occurs, or are the 3-hourly observations too infrequent?

The correlation analysis among the ABL height definitions is quite nice. I was surprised there was not a similar correlation analysis between the surface temperature and the ABL hight (around lines 427-455). In particular with regard to the seasonal variation that is mentioned, it would be nice to see whether the ABL height is related to the, say, the absolute maximum temperature or the diurnal temperature range.

Several places in the text seem to indicate that the presence of clouds might alter the evening transition (ET). This was never made completely clear. Is there a relationship or not? If there is, can it be understood in terms of the longwave effect that is mentioned, or is the mechanism unclear?

Technical Comments

Line 32: Start the sentence with "The"

Line 36: change to "balance between the surface"

Line 37-38: change to "The ABL height is a key parameter, providing a length scale for..."

Line 51: insert a dash ("–") between maintenance and rather

Line 54: delete the extra "m"

Line 66: I think there are many more studies of the diurnal variation of ABL height than this sentence would lead the reader to believe. There are recent examples using ARM sites (Santanello et al, 2007, http://dx.doi.org/10.1175/JHM614.1; May et al., 2012, http://dx.doi.org/10.1175/JCLI-D-11-00538.1), but there are also older

examples from field studies (Brill & Albrecht, 1982, http://dx.doi.org/10.1175/1520-0493(1982)110<0601:DVOTTW>2.0.CO;2) or observation sites (Hashiguchi et al., 1995a, Boundary-LayerMeteorology 74: 419-424; Hashiguchi et al., 1995b, http://dx.doi.org/10.1029/95RS00653), and even in more exotic settings (e.g., on a glacier, van den Broeke, 1997, Boundary-Layer Meteorology 83: 183–205).

Line 72: Also see Seidel et al. (2012, http://dx.doi.org/10.1029/2012JD018143).

Line 90: change to "... days in each month..."

Line 100: delete "continuously"

Line 106: delete "at"

Line 184: delete "convective"

Line 192: insert "as" before easy

Line 231: change "is" to "are"

Line 239: Doesn't this ABL structure seem similar to a shallow cumulus profile, or a decoupled cloud-topped ABL, as is often described over the ocean in the transition from stratocumulus to cumulus?

Line 291-292: change to "... ET process was not delayed and ..."

Line 292: I think this should read "On the third night the SBL was detected at a height near 0.45 km."

Line 298: Delete "till"

Figure 6b: This bar chart is difficult to read, the format in Figure 9 is much better.

---

## Referee Comment (RC2) · Anonymous Referee #2 · 12 Sep 2016

Review of the article titled "Diurnal variability of the atmospheric boundary layer height over a tropical station in the Indian monsoon region" by Mehta and coauthors for publication in the Atmospheric Chemistry Physics.

The authors have used data collected by the radiosondes over a tropical station and deduced the boundary layer height. The data were collected over 3-year period during various field campaigns. They have shown the diurnal, and seasonal cycle of boundary layer depth. Further they have classified the boundary layer structure into different categories like convective, stable and residual and have reported the statistics of those as well. The authors have made a good attempt to report the statistics but they fall short in deriving any scientific conclusions from them, leaving the reader with a feeling

that no manuscript is simply a collection of statistics. I suggest the manuscript to go through a thorough revision before being published. Below I have listed my major and minor concerns.

Major Concerns: 1) As I mentioned earlier, the paper seems like a collection of statistics. You have mentioned in the abstract that various studies have reported the boundary layer depth from that station. So I am not sure of the purpose of this paper is to validate them, or to report them again or to gain some scientific insights on the causes of the changes in the boundary layer depth. It will be good if you can clarify it in the introduction section. 2) As you have radiosonde data, I suggest you calculate the lifting condensation level (LCL) and also report its variation for the different boundary layers. Please refer to Bolton (1980) regarding the calculations. Add the LCL to Figure 8 and 10. 3) You can calculate the equivalent potential temperature and saturation equivalent potential temperature from Bolton (1980) and then further calculate the convective available potential energy (CAPE) and Convective Inhibition (CINE). These are very important quantities and will make the article very robust. 4) You have reported the Cloud top heights (CTH) from the satellite measured TBB. It will be great if you report the cloud base height and cloud top heights from the radiosodes themselves. The RH measurements will tell you when the sensor is passing through cloud layers. The derived cloud base height then can be added to figure 8 and 10. You can then classify the thermodynamic structure based on cloud thickness rather than cloud top heights. 5) You have made a very good attempt at classifying the BL structure as convective+residual, stable, stable+convective etc. It will be very nice if you can make a cartoon similar to Figure 9.21 of Wallace and Hobbs book with actual values you have for the summer and winter seasons. Thanks.

Minor concerns: 1) The shades are not visible in the Table. 2) Line 15: Please add MSL after lat, lon 3) Line 22: I would say "constant" rather than "steady". 4) Line 36: You mean Stull 1988 not 1998. 5) Line 39: You mean to say "convective" and not "convection" 6) Line 45-60: what about the role of shear and radiation. 7) Line 65-70:

Might be good to refer to Schmidt and Niyogi. 8) Line 74: You mean to say "remote sensing" not "remote sounding". 9) Line 90: "launches" and not "launchings"/ 10) line 92: "has" and not "have"/ 11) Line 97: Please list the full-form of the acronum CAWSES 12) Line 165: It might be good to mention that the reported drift is below 4km. 13) Line 425-426: Please rephrase. "Attains" is misleading. 14) Figure 3 legend is incorrect. 15) Figure 4: I believe you have listed the lines for sunset and sunrise backwards. 16) Figure 6a: Why do you have two black bars surrounding the yellow bars.

---

## Author Response (AR1)

**Response to Referee #1**

Thank you very much for reviewing our manuscript and providing potential comments.

*General Comment: This study presents an analysis of 3-hourly radiosonde data taken over 3-day intervals in each month over several years. The focus is on identifying the boundary layer height. This height is divided into convective and stable boundary layers and also residual layers. Several methods are used to identify the top of the ABL, but they are all based on gradients. These methods are appropriate, and the comparison of the methods is a nice aspect of this study. The diurnal cycle of ABL height is described; the upshot seems to be that there is a strong diurnal cycle but the amplitude is affected by the season as well as the presence of clouds, which also appear to affect the phase. Overall, the presentation of the results is clear, but the weakness is that it is not clear whether the results provide novel insight. Instead, the novelty appears to be in the data set itself. There are several areas where some additional work could improve the analysis. These are relatively minor issues. The text is written well, but could stand another round of close editing for small grammatical and English issues (some examples listed below).*

Reply: We have taken all the suggestions and incorporated into the revised manuscript.

Comment 1: *Perhaps the main issue I have with this study is that it is quite focused on the radiosondes, with limited support from other observations. This becomes crucial as the text explores the impact of clouds on the ABL structure. There is a good use of IR brightness temperature to provide an estimate of cloud top height, but this is the only cloud observations that are presented. I found that to be surprising. Perhaps even more surprising once I visited the NARL website (https://www.narl.gov.in/) and found that there are several instruments that could provide useful supplementary data. One that could provide highly complementary data is the microwave radiometer; the web site says that it retrieves cloud base and liquid water path. These could be quite useful for more clearly defining the cloud layer. There are also radiation sensors and eddy covariance latent and sensible heat fluxes that could be used to construct a surface energy balance. There are also rain gauges and a disdrometer, which could be used to explore the ABL height as a function of rain rate. Such an analysis could bolster the conclusions about deep convection having little impact on the ABL height.*

**Reply 1:** Certainly appreciate the reviewer's suggestion to utilize the microwave radiometer data for the cloud layer information and boundary layer tower data to calculate the surface energy balance. Unfortunately, these datasets is not available during observation periods used in this study.  Reviewer 2 also pointed to obtain the cloud layer using relative humidity (RH) data (See response to reviewer#2). We have checked the rainfall data obtained from rain gauges (Automatic weather station data), but we could not find any relation between the ABL and rainfall.

**Changes in the manuscript 1:**
**(~Line 146-148)** Most of the observations are conducted during non-rainy days except two during 01:00 IST-02:00 IST on 18 August 2011 and 14:00 IST-20:00 IST on 21 August 2012, with total rainfall about 47 mm and 46 mm, respectively.

**(~Line 630)** It is to be noted that there are two occasions when rainfall occurred during the campaign periods, however, these few data do not reveal any relation between the ABL and rainfall.

Comment 2: *The histograms of Figure 7 raise an issue about the statistics being used. Most of the histograms (which are NOT pdfs) look very non-Gaussian. The text mentions that the peak of the SBL histogram is at a substantially lower altitude than the mean (also true for RL in summer). Based on these histograms, I suggest also reporting the median and interquartile range, which provide a better estimate of the typical values and variability of the data.*

**Reply 2:** The distribution shown in Fig. 7 is non-Guassian especially for the SBL distribution which has longer tail on the right and hence positively skewed.  As suggested, we have also included the boxplot of

the SBL, CBL and RL heights for the annual, winter and summer monsoon as shown in Fig. 8 in the revised manuscript.

**Changes in the manuscript 2:**
(**~Line 468-475**) We also obtained the distribution of the SBL, CBL and RL for the annual, winter and summer monsoon in terms of the boxplot as shown in the Figs. 8a-c, respectively. The median values the SBL during the annual, winter and summer remain same (Fig. 8a). There are a few outliers whose values are greater than 3 times of the corresponding interquartile ranges (IQR) for the annual and two different seasons. The SBL mostly lies below 0.65 km during the winter and 0.4 km during summer monsoon. The SBL is more variable during the winter than summer monsoon. As mentioned earlier, the CBL is higher and more variable during the summer monsoon than winter (Fig. 8b). Similarly, the RL is also more variable during the summer monsoon than winter (Fig. 8c). In contrast to the CBL, RL is lower during the summer monsoon when compared to winter.

*Comment 3: I found the definition of the residual layer (RL) to be a little unclear. It seems to be defined exactly the same as for the CBL, is that correct? It would be good to include an explanation in Section 2.3.*

**Reply 3**: The method to obtain the RL is similar to that of the CBL. As the RL is the part of the daytime ABL, its characteristic is entirely similar to that of CBL except RL does not connected to surface whenever SBL is present. However, for the case when the SBL is absent RL is connect to the surface and generally referred as neutral RL (NRL) (Liu and Liang, 2010). However, we have referred it as simply RL. This aspect has been mentioned in the manuscript.
Liu, S. and Liang, X.-Z.: Observed diurnal cycle climatology of planetary boundary layer height, J. Climate, 23, 5790-5809, 2010.

**Changes in the manuscript 3:**
(**~Line 193**) The method to obtain the RL is similar to that of the CBL.

*Comment 4: One aspect of the residual layer that has been pointed out as being important for the diurnal evolution of the ABL is that it provides the potential for "explosive growth" of the ABL as a CBL forms in the morning and grows into the RL. This was not mentioned in this paper. Is it possible to quantify whether this explosive growth occurs, or are the 3-hourly observations too infrequent?*

**Reply 4:** This is a good point. We have included this information in the revised manuscript. One can simply obtain the difference between ABL height before and after the sunrise, in order quantify the CBL growth. However, from typical diurnal variations (See Fig. 4), it can be seen that the transition from RL to CBL is not always explosive.

**Changes in the manuscript 4:**
(**~Line 60-62**) One aspect of the RL that has been pointed out as being important for the diurnal evolution of the ABL is that it provides the potential for "explosive growth" of the ABL as a CBL forms in the morning and grows into the RL.

*Comment 5: The correlation analysis among the ABL height definitions is quite nice. I was surprised there was not a similar correlation analysis between the surface temperature and the ABL height (around lines 427-455). In particular with regard to the seasonal variation that is mentioned, it would be nice to see whether the ABL height is related to the, say, the absolute maximum temperature or the diurnal temperature range.*

**Reply 5:** We have obtained the scatter plot of the surface temperature and ABL height. The scatter diagrams of the surface temperature and the different ABL regimes such as the CBL, RL and SBL indicate that their relations are random in nature. Though CBL and RL become higher with higher surface temperature and vice versa, there are several occasions when they vary randomly.

**Changes in the manuscript 5:**
**(~Line 547-553)** As mentioned earlier, the diurnal evolution of the annual and seasonal mean pattern of the ABL is closely associated with the surface temperature. In order to see their 3-hourly relationships, we obtained the scatter plot of the CBL, RL and SBL with the surface temperature as shown in Figs. 11a-c, respectively. Broadly, the scatter diagram indicates that warmer is the surface, the higher is the CBL and RL and vice versa (Figs. 11a-b). However, these features are not always consistent and several times they occur randomly. In contrast to the CBL and RL, SBL higher is higher over the colder surface and vice versa, however, these features also are not always consistent and several times they occur randomly (Fig. 11c).

*Comment 6: Several places in the text seem to indicate that the presence of clouds might alter the evening transition (ET). This was never made completely clear. Is there a relationship or not? If there is, can it be understood in terms of the longwave effect that is mentioned, or is the mechanism unclear?*

**Reply 6:** The presence of the cloud could be one possible reason to alter the evening transition. However, there are occasions when the SBL forms even in the presence of the clouds (See Table 1). So at this juncture, it is not possible to provide a clear mechanism only based on the cloud information.

**Changes in the manuscript 6:**
**(~Line 355-358)** However, it must be noted that presence of the clouds could be one possible reason as there as occasions when the SBL forms even in the presence of the clouds. Hence, the delay in the ET process cannot be entirely explained by the longwave effect.

*Technical Comments*

*Line 32: Start the sentence with "The"*
Reply: Corrected.

*Line 36: change to "balance between the surface"*
Reply: Changed.

*Line 37-38: change to "The ABL height is a key parameter, providing a length scale for..."*
Reply: Changed.

*Line 51: insert a dash ("–") between maintenance and rather*
Reply: Inserted.

*Line 54: delete the extra "m"*
Reply: Changed.

*Line 66: I think there are many more studies of the diurnal variation of ABL height than this sentence would lead the reader to believe. There are recent examples using ARM sites (Santanello et al, 2007, http://dx.doi.org/10.1175/JHM614.1; May et al., 2012, http://dx.doi.org/10.1175/JCLI-D-11-00538.1), but there are also older examples from field studies (Brill & Albrecht, 1982, http://dx.doi.org/10.1175/1520-0493(1982)110<0601:DVOTTW>2.0.CO;2) or observation sites (Hashiguchi et al., 1995a, Boundary-LayerMeteorology 74: 419-424; Hashiguchi et al., 1995b, http://dx.doi.org/10.1029/95RS00653), and even in more exotic settings (e.g., on a glacier, van den Broeke, 1997, Boundary-Layer Meteorology 83: 183–205).*
Reply: We thank for updating these references which are cited in the revised manuscript.

**Changes in the manuscript:**
(~Line 86-104) There are several case studies focusing on the diurnal structure of the boundary layer and the mechanism responsible for its formation over different regions of the globe. Brill and Albrecht, (1982) presented the diurnal variation of the cloud fraction and trade-wind inversion base height using the data collected from various ships and aircraft. May et al., (2012) have studied the diurnal variation of convection, cloud, radiation, and boundary layer structure in the coastal monsoon environment (Darwin, Australia). Santanello et al., (2007) has studied the feedback of soil moisture dryness on the development of the convective boundary layer over the southern Atmospheric Research Measurement Program–Great Plains Cloud and Radiation Testbed (ARM-CART) sites. During a clear sky day Hashiguchi et al., (1995a) observed boundary layer radar echo indicating the ABL height. Hashiguchi et al., (1995b) further observed that the boundary layer radar detects diurnal variation of the ABL both at the equatorial and midlatitude regions. During the summer over a midlatitude glacier, the diurnal and vertical and horizontal structures of the boundary layer were found to be dominated by persistent glacier winds forced by gravity (Van den Broeke, 1997).

*Line 72: Also see Seidel et al. (2012, http://dx.doi.org/10.1029/2012JD018143).*
Reply: Cited.

*Line 90: change to "... days in each month..."*
Reply: Changed.

*Line 100: delete "continuously"*
Reply: Deleted.

*Line 106: delete "at"*
Reply: Deleted.

*Line 184: delete "convective"*
Reply: Deleted.

*Line 192: insert "as" before easy*
Reply: Inserted.

*Line 231: change "is" to "are"*
Reply: Changed.

*Line 239: Doesn't this ABL structure seem similar to a shallow cumulus profile, or a*
*decoupled cloud-topped ABL, as is often described over the ocean in the transition*
*from stratocumulus to cumulus?*
Reply:  Yes, it seems like a shallow layer cloud decoupled from the surface. The surface moisture is very small and the LCL is observed at about 0.6 km and CTH is at about 0.81 km indicating that a shallow layer cloud of thickness about 0.2 km decoupled from the surface.

**Changes in the manuscript:**
(~Line 291-293) The surface moisture is very small and the LCL observed at about 0.6 km which indicates that a shallow layer cloud of thickness about 0.2 km decoupled from the surface (Garratt, 1992).

*Line 291-292: change to "... ET process was not delayed and ..."*
Reply: Changed.

*Line 292: I think this should read "On the third night the SBL was detected at a height*
*near 0.45 km."*
Reply: Modified as suggested.

*Line 298: Delete "till"*
Reply: Deleted.

*Figure 6b: This bar chart is difficult to read, the format in Figure 9 is much better.*
Reply: Figure 6b is modified as Figure 9.

**Response to Referee #2**
Thank you very much for reviewing our manuscript and providing potential comments.

*General comment: The authors have used data collected by the radiosondes over a tropical station and deduced the boundary layer height. The data were collected over 3-year period during various field campaigns. They have shown the diurnal, and seasonal cycle of boundary layer depth. Further they have classified the boundary layer structure into different categories like convective, stable and residual and have reported the statistics of those as well. The authors have made a good attempt to report the statistics but they fall short in deriving any scientific conclusions from them, leaving the reader with a feeling that no manuscript is simply a collection of statistics. I suggest the manuscript to go through a thorough revision before being published. Below I have listed my major and minor concerns.*

**Reply:** We have taken all these suggestions and incorporated into the revised manuscript.

*Major Concerns:*

*Comment1) As I mentioned earlier, the paper seems like a collection of statistics. You have mentioned in the abstract that various studies have reported the boundary layer depth from that station. So I am not sure of the purpose of this paper is to validate them, or to report them again or to gain some scientific insights on the causes of the changes in the boundary layer depth. It will be good if you can clarify it in the introduction section.*

**Reply 1:** There have been several case studies over the station, but none of them has classified the ABL into different regimes such as CBL, SBL and RL and has not been dealt separately during different seasons. The effect of the cloud on the diurnal structure of the ABL is also, has not been attempted yet before this study.

**Changes in manuscript** 1
**(~ Line 122-125)** Making use of this dataset (radiosonde data), first time complete diurnal variability of the ABL height and their classification into different ABL regimes such as the CBL, SBL and RL during different seasons and effect of the cloud in its diurnal structure has been studied and the results are presented in this paper.

*Comment 2) As you have radiosonde data, I suggest you calculate the lifting condensation level (LCL) and also report its variation for the different boundary layers. Please refer to Bolton (1980) regarding the calculations. Add the LCL to Figure 8 and 10.*

**Reply 2:** We have added the lifting condensation level (LCL) in the revised manuscript.

**Changes in manuscript 2:**
**(~Line 203-211)** The lifting condensation level (LCL) is defined as the height at which an unsaturated air parcel becomes saturated (RH >100%) when it is cooled by dry adiabatic lifting (Wallace and Hobbs, 2006). It provides an empirical estimate of the cloud base height. The temperature ($T_L$) pressure ($P_L$) and height ($Z_L$) of the LCL is obtained using following equations (Bolton 1980; Stull 1988; Anurose et al., 2016):

$$T_L = \frac{2840}{3.5 \ln(T_{30m}) - \ln(Pw_{30m}) - 4.805} + 55 \qquad (1)$$

$$P_L = P_{30m} \left[\frac{T_L}{T_{30m}}\right]^{3.5} \qquad (2)$$

$$Z_L = -H \ln(P_L/P_0) \qquad\qquad\qquad (3)$$

where $T_{30m}$, $P_{30m}$ and $Pw_{30m}$ are temperature, pressure and water vapor pressure at 30 m height, respectively, $P_0$ is surface pressure and $H$ is scale height taken as 7.5 km (Wallace and Hobbs 2006).

**(~Line 555-575)** The LCL generally occurs either below or at the CBL and RL except a few times when it occurs above the CBL and RL (Figs. 11d-e). The cases when the LCL occurs above the CBL or RL, clouds may not be generated by the processes driven by the ABL and can be formed due to large scale-dynamics (Anurose et al., 2016). We observed no relationship between the SBL and LCL (Fig. 11f). For the SBL case, as the vertical motion is inhibited, the relationship between the LCL and SBL is irrelevant (Anurose et al., 2016). Anurose et al., (2016) also studied the relationship between the CBL height and the LCL over the coastal station, Thiruvananthapuram (8.5◦ N, 76.9◦ E), they did not observed any relationship. However, the LCL over Thiruvananthapuram is found to higher than ABL for a majority of the database in contrast to Gadanki.

**(~Line 620-629)** When the CTH is within ± 0.5 km of the $ABL_{CR}$, LCL occurs mostly below the ABL, except a few cases when it coincides with either the ABL or CTH (Fig.13a). When the CTH is below 0.5 km of the $ABL_{CR}$, the LCL again occurs mostly below the ABL but generally coincided with the CTH (Fig.13b). In this case, the LCL sometimes also occurs above the CTH. The clouds occurring below the ABL could be the shallow clouds, in such cases LCL representing the cloud base may occur near to the CTH. However, it is to be noted that the CTH represents the cloud condition for the area averaged over $0.25^o$ latitude X $0.25^o$ longitude regions, whereas the LCL indicates the cloud base exactly over the observation site. Thus, the LCL may not always agree with the CTH when the cloud is not extended over the larger area. For the cases when the clouds occurring above 0.5 km but below 6.0 km of the $ABL_{CR}$, the LCL mostly occurs either below the ABL or generally coincides with the ABL (Fig 13c).

*Comment 3) You can calculate the equivalent potential temperature and saturation equivalent potential temperature from Bolton (1980) and then further calculate the convective available potential energy (CAPE) and Convective Inhibition (CINE). These are very important quantities and will make the article very robust.*

**Reply 3**: Certainly appreciate the helpful suggestion; however, carrying out this is beyond the scope of the present work. We will take up these works in the future as follow-up studies.

**Changes in manuscript 3**: No changes made in the revised manuscript.

*Comment 4) You have reported the Cloud top heights (CTH) from the satellite measured TBB. It will be great if you report the cloud base height and cloud top heights from the radiosondes themselves. The RH measurements will tell you when the sensor is passing through cloud layers. The derived cloud base height then can be added to figure 8 and 10. You can then classify the thermodynamic structure based on cloud thickness rather than cloud top heights.*

**Reply 4:** Following the method of Wang and Rassow (1995), the cloud base height is obtained from the relative humidity (RH) measurement as shown in the Figure below. The time series of the cloud

base height whenever detected using RH in general agree with the cloud top height obtained using TBB data except during deep convection events. However, as the criteria for fixing the cloud base and top heights using the RH data has not been finalized for the tropical clouds occurring over this region, we have preferred satellite derived CTH data in this study. For instance, during December 18-21, 2013, when deep convection occurred, RH has never exceeded 60% and indicates no cloud based on the RH measurement. The detailed study pertaining to the identification of the vertical structure of cloud using RH and its comparison with one identified using satellite brightness temperature data we plan to carry out in the future.

[Figure]

Figure: Time series of the Cloud base height obtained from RH measurement and Cloud top height obtained using satellite measurements.

Wang, J. and Rossow, W.B.: Determination of cloud vertical structure from upper-air observations. Journal of Applied Meteorology, 34(10), pp.2243-2258, 1995.

**Changes in manuscript 4:**

**(~Line 709-711)** As suggested by Wang and Rassow (1995), the various cloud layers can be obtained utilizing the RH data (Wang and Rassow, 1995). However, as the criteria for fixing the cloud base and top heights using the RH data has not been finalized for the tropical clouds occurring over this region, we have preferred satellite derived CTH data in this study.

*Comment 5) You have made a very good attempt at classifying the BL structure as convective+residual, stable, stable+convective etc. It will be very nice if you can make a cartoon similar to Figure 9.21 of Wallace and Hobbs book with actual values you have for the summer and winter seasons. Thanks.*

**Reply 5**: Thanks for nice suggestion. We have added a cartoon representing the diurnal evolution of the ABL during the summer and winter seasons similar to Wallace and Hobbs (2006).

Wallace J.M., Hobbs, P.V.: Atmospheric science an introductory survey, second edition. International Geophysics series, Acadamic Press 92, 483 pp, 2006.

**Changes in manuscript 5:**

(**~Line 576-592**) Figures 12a and 12b show the schematic representation of the diurnal evolution of the mean ABL from 11:00 IST on the first day to 11:00 IST on the second day during the winter and summer monsoon seasons, respectively. The diagram is generated from the seasonal mean ABL height data presented in Figs.10a and 10b. The vertical cross section of the ABL characterizing the seasonal mean CBL, SBL, RL, and entrainment zone (E.Z.), capping inversion and the LCL obtained from the observed data over Gadanki. The schematic diagram represents the typical evolution of the boundary layer, consistent to the diagram presented in the Stull (1988) and Wallace and Hobbs (2006). The CBL during the winter evolves slowly when compared to summer monsoon season in which the ABL growth is rapid. The SBL starts to form well before the sunset during both seasons; however, it remains persistent, even after the sunrise only during the winter season. During the winter, the RL remains almost constant throughout the night. However, during the summer, the RL rapidly decreases as the night passes. The capping inversion during the summer is thicker when compared to the summer monsoon. We have also shown the seasonal mean LCL which occurs within the CBL and RL during both seasons. Note that the transition regions (from the CBL to RL during evening transition and the RL to CBL during morning transition) cannot be accurately represented with available time resolutions. Thus, the part of the CBL after the sunset and the part of the RL after the sunrise may not possess any meaning. This schematic diagram clearly represents the typical (Fig.4), annual mean (Fig.9) and seasonal mean (Fig.10) characteristic of the ABL.

*Minor concerns:*

*1) The shades are not visible in the Table.*

Reply 1: Changed to italic font along with shades.

*2) Line 15: Please add MSL after lat, lon*

Reply2: Added.

*3) Line 22: I would say "constant" rather than "steady".*

Reply 3: Changed.

*4) Line 36: You mean Stull 1988 not 1998.*

Reply 4: Corrected.

*5) Line 39: You mean to say "convective" and not "convection"*

Reply 5: Changed.

*6) Line 45-60: what about the role of shear and radiation.*

Reply 6: Added to L59 in the revised manuscript.

*7) Line 65-70: Might be good to refer to Schmidt and Niyogi.*

Reply 7: Cited.

*8) Line 74: You mean to say "remote sensing" not "remote sounding".*

Reply 8: Corrected.

*9) Line 90: "launches" and not "launchings"*

Reply 9: Corrected.

*10) line92: "has" and not "have"*

Reply 10: Corrected.

*11) Line 97: Please list the full-form of the acronum CAWSES*

Reply 11: Added.

*12) Line 165: It might be good to mention that the reported drift is below 4km.*

Reply 12: Mentioned.

*13) Line 425-426: Please rephrase. "Attains" is misleading.*

Reply 13: Changed.

*14) Figure 3 legend is incorrect.*

Reply14: Corrected.

*15) Figure 4: I believe you have listed the lines for sunset and sunrise backwards.*

Reply 15: Yes, Corrected.

*16)Figure 6a: Why do you have two black bars surrounding the yellow bars.*

Reply 16: As black bar is wider than yellow bar, appears as two black bars. Black bar is kept back of yellow bar. As suggested by reviewer# 1, we have modified figure 6a as figure 10 in the revised manuscript.

[revised manuscript text omitted]

---

## Author Response (AR2)

**To**
**Dr. Hailong Wang**
**Co-Editor, Atmospheric Chemistry and Physics**

**Response to the Co-Editor**
This is the first revision of the manuscript. Authors have incorporated the suggestions made by both reviewers and it has resulted in substantial improvement in the manuscript. However, I think the authors have made mistakes in calculating some of the parameters and I think the manuscript needs to be revised again to correct those.

Thank you very much for monitoring our manuscript and providing positive feedbacks to improve the manuscript. I have incorporated all the suggestions by the Reviewer#2 in the revised manuscript. I would like to thank Reviwer#2 for reviewing our manuscript again and providing potential comments in order to improve the manuscript further.

*Comment 1. Calculation of the LCL height (Zlcl) is incorrect. The Bolton (1980) formulation gives you the temperature of the LCL, to calculate the height simply subtract the surface air temperature and divide by the dry adiabatic lapse rate (g/Cp=9.8). There is no need to invoke pressure and scale height etc. in this calculation. The reported values of Zlcl are a lot higher than expected.*

*Zlcl = (Tlcl-Tsfc)/(g/Cp).*

**Reply 1:** Based on the Reviewer's suggestion we have modified the calculation for the LCL height (Zlcl1). However, we would like to point out that the calculation based on the equation (1) (as suggested by reviewer) and the one (Zlcl2) using equations (2) and (3) (in our manuscript) are nearly the same as shown in figure (1). From figure 1 it can be seen that that the difference between Zlcl1 and Zlcl2 is very small. But we also agree with the Reviewer that the calculation of Zlcl2 depends upon the scale height, while calculation of Zlcl1 is straight forward. Hence, we have revised the formula to calculate the LCL height in the revised manuscript.

$$Z_{lcl1} = -(T_{lcl} - T_{sfc})/(\frac{g}{C_p}) \qquad (1)$$

$$P_{lcl} = P_{30m} \left[\frac{T_{lcl}}{T_{sfc}}\right]^{3.5} \qquad (2)$$

$$Z_{lcl2} = -H \ln(P_{lcl}/P_{sfc}) \qquad (3)$$

[Figure]

**Figure 1: Time series of the $Z_{lcl1}$, $Z_{lcl2}$, and difference between $Z_{lcl1}$ -$Z_{lcl2}$.**

**Changes in manuscript 1**: (~Line 177-179)
$$Z_L = -(T_L - T_{30m})/\tau_d \qquad\qquad (2)$$

where $T_{30m}$ and $P_{w30m}$ are temperature and water vapor pressure at 30 m height, respectively, $\tau_d$ is dry adiabatic lapse rate.

*Comment 2. If the LCL is lower than the inversion base height and the boundary layer is convective then clouds form. As you have the cloud boundaries (base and top) for each campaign, please include them in the schematic in Fig 12.*

**Reply 2:** We have already provided the LCL height which represents the base of clouds. In the revised manuscript, we have added cloud top height (CTH) height based on the TBB data.

**Changes in manuscript 2**: (~L520-525)
We have also shown the seasonal mean LCL and CTH representing the cloud base and top, respectively. The LCL occurs within the CBL and RL during both the seasons. The mean CTH has very little variation over the diurnal cycle during both the seasons. During winter season, the CTH is slightly higher during morning hours between 0500 IST to 08000 IST. It could be due to frequent occurrence of the fogs or stratus clouds in the morning hours during winter season over Gadanki region.

**Few Minor issues:**

*1. The text can go through some improvements as many portions of the introduction and summary section are not related to the current research and could be trimmed. For example Line 63-100, I am not sure how studies conducted over glaciers (line 85) are relevant here. In the revised version, you have modified the text to include the specific objectives of this study. However, I highly encourage the authors to consider mentioning the objectives in the following way "The objectives of this study are 1)*

*document the boundary layer heights from radiosonde data and 2) study the evolution of the different layers within the ABL". This is similar to the way you have reported the conclusions.*

**Reply 1:** We have revised the introduction section and modified based on the reviewer's suggestions. As focus of this study to document of the diurnal variation of the ABL over the Indian monsoon region, those literatures describe the diurnal evolution of the ABL over different meteorological conditions across different regions of the globe has been cited in the introduction.

**Changes in the manuscript:**
**(~L104-107)** The objectives of present study are to (i) document the boundary layer heights from radiosonde data, (ii) study the evolution of the different ABL regimes such as the CBL, SBL and RL, and (iii) qualitatively describe the nature of the ABL in the presence of clouds.

*2. Is the IST similar to local time over Gadanki?*

**Reply 2:** Yes, local time and IST over Gadanki are the same.

*3. Caption of Figure 5: please change to "scatter plot between .." from "correlation". The caption is meant to describe the figures and not its objectives.*

**Reply 3:** Changed.

*4. There are several small typos in the manuscript as is, I recommend the authors to do a thorough grammatical revision of it.*

**Reply 4:** We have gone through the manuscript thoroughly and corrected the grammatical mistakes in the revised manuscript.

**Response to Referee #2**

Thank you very much for reviewing our manuscript again and providing potential comments in order to improve the manuscript further.

*Comment 1. Calculation of the LCL height (Zlcl) is incorrect. The Bolton (1980) formulation gives you the temperature of the LCL, to calculate the height simply subtract the surface air temperature and divide by the dry adiabatic lapse rate (g/Cp=9.8). There is no need to invoke pressure and scale height etc. in this calculation. The reported values of Zlcl are a lot higher than expected.*

*Zlcl = (Tlcl-Tsfc)/(g/Cp).*

**Reply 1:** Based on the Reviewer's suggestion we have modified the calculation for the LCL height (Zlcl1). However, we would like to point out that the calculation based on the equation (1) (as suggested by reviewer) and the one (Zlcl2) using equations (2) and (3) (in our manuscript) are mostly similar as shown in figure (1). From figure 1 it can be seen that that the difference between Zlcl1 and Zlcl2 is very small. But we agree with the reviewer that the calculation of Zlcl2 depends upon the scale height, while calculation of Zlcl1 is straight forward. Hence, we have revised the formula to calculate the LCL height in the revised manuscript.

$$Z_{lcl1} = -(T_{lcl} - T_{sfc})/(\frac{g}{C_p}) \qquad (1)$$

$$P_{lcl} = P_{30m} \left[\frac{T_{lcl}}{T_{sfc}}\right]^{3.5} \qquad (2)$$

$$Z_{lcl2} = -H \ln(P_{lcl}/P_{sfc}) \qquad (3)$$

[Figure]

**Figure 1: Time series of the $Z_{lcl1}$, $Z_{lcl2}$, and difference between $Z_{lcl1}$ -$Z_{lcl2}$.**

**Changes in manuscript 1**: (~Line 177-179)

$$Z_L = -(T_L - T_{30m})/\tau_d \tag{2}$$

where $T_{30m}$ and $P_{w30m}$ are temperature and water vapor pressure at 30 m height, respectively, $\tau_d$ is dry adiabatic lapse rate.

*Comment 2. If the LCL is lower than the inversion base height and the boundary layer is convective then clouds form. As you have the cloud boundaries (base and top) for each campaign, please include them in the schematic in Fig 12.*

**Reply 2:** We have already provided the LCL height which represents the base of clouds. In the revised manuscript, we have CTH height based on the TBB data.

**Changes in manuscript 2**:

**Few Minor issues:**

*1. The text can go through some improvements as many portions of the introduction and summary section are not related to the current research and could be trimmed. For example Line 63-100, I am not sure how studies conducted over glaciers (line 85) are relevant here. In the revised version, you have modified the text to include the specific objectives of this study. However, I highly encourage the authors to consider mentioning the objectives in the following way "The objectives of this study are 1) document the boundary layer heights from radiosonde data and 2) study the evolution of the different layers within the ABL". This is similar to the way you have reported the conclusions.*

**Reply 1:** We have revised the introduction section and modified based on the reviewer's suggestions. As focus of this study to document of the diurnal variation of the ABL over the Indian monsoon region, those literatures describe the diurnal evolution of the ABL over different meteorological conditions across different regions of the globe has been cited in the introduction.

**Changes in the manuscript:**
**(~L104-107)**  The objectives of present study are to (i) document the boundary layer heights from radiosonde data, (ii) study the evolution of the different ABL regimes such as the CBL, SBL and RL, and (iii) qualitatively describe the nature of the ABL in the presence of clouds.

*2. Is the IST similar to local time over Gadanki?*

**Reply 2:** Yes, local time and IST over Gadanki is the same.

*3. Caption of Figure 5: please change to "scatter plot between .." from "correlation". The caption is meant to describe the figures and not its objectives.*

**Reply 3:** Changed.

*4. There are several small typos in the manuscript as is, I recommend the authors to do a thorough grammatical revision of it.*

**Reply 4:** We have gone through the manuscript thoroughly and corrected the grammatical mistakes in the revised manuscript.

[revised manuscript text omitted]